# Neural tube-associated boundary caps are a major source of mural cells in the skin

**Gaspard Gerschenfeld**[1,2†], **Fanny Coulpier**[1,3,4†], **Aurélie Gresset**[1], **Pernelle Pulh**[1,3], **Bastien Job**[5], **Thomas Topilko**[6], **Julie Siegenthaler**[7], **Maria Eleni Kastriti**[8,9], **Isabelle Brunet**[10], **Patrick Charnay**[1], **Piotr Topilko**[1,3]*

[1]Institut de Biologie de l'Ecole normale supérieure (IBENS), Ecole normale supérieure, CNRS, Inserm, Université PSL, Paris, France; [2]Sorbonne Université, Collège Doctoral, Paris, France; [3]nstitut Mondor de Recherche Biomédicale, Inserm U955-Team 9, Créteil, France; [4]Genomic facility, Ecole normale supérieure, PSL Research University, CNRS, Inserm, Institut de Biologie de l'Ecole normale supérieure (IBENS), Paris, France; [5]Inserm US23, AMMICA, Institut Gustave Roussy, Villejuif, France; [6]Laboratoire de Plasticité Structurale, Sorbonne Université, ICM Institut du Cerveau et de la Moelle Epinière, Inserm U1127, CNRS UMR7225, Paris, France; [7]Department of Pediatrics Section of Developmental Biology, University of Colorado Anschutz Medical Campus, Aurora, United States; [8]Department of Physiology and Pharmacology, Karolinska Institutet, Stockholm, Sweden; [9]Department of Neuroimmunology, Center for Brain Research, Medical University Vienna, Vienna, Austria; [10]Inserm U1050, Centre Interdisciplinaire de Recherche en Biologie (CIRB), Collège de France, Paris, France

**\*For correspondence:**
piotr.topilko@inserm.fr

[†]These authors contributed equally to this work

**Competing interest:** The authors declare that no competing interests exist.

**Abstract** In addition to their roles in protecting nerves and increasing conduction velocity, peripheral glia plays key functions in blood vessel development by secreting molecules governing arteries alignment and maturation with nerves. Here, we show in mice that a specific, nerve-attached cell population, derived from boundary caps (BCs), constitutes a major source of mural cells for the developing skin vasculature. Using Cre-based reporter cell tracing and single-cell transcriptomics, we show that BC derivatives migrate into the skin along the nerves, detach from them, and differentiate into pericytes and vascular smooth muscle cells. Genetic ablation of this population affects the organization of the skin vascular network. Our results reveal the heterogeneity and extended potential of the BC population in mice, which gives rise to mural cells, in addition to previously described neurons, Schwann cells, and melanocytes. Finally, our results suggest that mural specification of BC derivatives takes place before their migration along nerves to the mouse skin.

## Editor's evaluation

The authors show that Krox20 positive boundary cap cells travel along the nerves to the dermis and become incorporated into the vascular plexus to form dermal mural cells, confirmed by a cluster in single cell RNA sequencing. This provides the first evidence of a boundary cap contribution to a portion of mural cells in the trunk dermis.

## Introduction

Blood vessels and nerves are branched structures that are established in parallel during development to supply almost every organ in the body. Their patterning is achieved through the coordinated action of a common set of factors (*Carmeliet and Tessier-Lavigne, 2005*). Indeed, recent studies have shown that in addition to using similar signals to differentiate, grow, and navigate, the vascular and nervous systems maintain a cross-talk that plays a key role in their mutual development. For instance, sympathetic nerves follow specific guidance cues produced by the vasculature to reach their appropriate targets (*Glebova and Ginty, 2005*). Reciprocally, studies by Mukouyama and colleagues have demonstrated that nerves and Schwann cell precursors (SCPs) play a decisive role in the development of the vascular network, as they provide essential diffusible factors, drive the alignment of blood vessels with nerves, and promote arteriogenesis (*Li et al., 2013*; *Mukouyama et al., 2005*; *Mukouyama et al., 2002*). Using *Neurogenin* mutant mice, which lack cutaneous nerves, they showed that both blood vessel patterning and arterial differentiation are prevented (*Mukouyama et al., 2002*). Furthermore, using in vitro and in vivo approaches, they demonstrated that two proteins secreted by cutaneous nerves, CXCL12 and VEGF-A, act as key factors to trigger the remodeling and nerve alignment of cutaneous blood vessels and arterial differentiation, respectively (*Li et al., 2013*; *Mukouyama et al., 2005*).

Over the last decade, numerous studies have revealed an unexpected plasticity of SCPs, which give rise, apart from a plethora of glial derivatives, to skin melanocytes, adrenal medulla chromaffin cells, tooth mesenchymal stem cells, osteocytes, chondrocytes, and enteric and parasympathetic neurons (*Adameyko et al., 2009*; *Dyachuk et al., 2014*; *Espinosa-Medina et al., 2014*; *Furlan et al., 2017*; *Kaucka et al., 2016*; *Uesaka et al., 2015*; *Xie et al., 2019*). However, a cellular contribution from SPCs to the endothelial or mural components of the vascular plexus has not been reported so far. Mural cells (MCs) include pericytes and vascular smooth muscle cells (vSMCs) that cover capillaries and larger vessels, respectively, and participate in blood vessel remodeling and stabilization (*Holm et al., 2018*). MCs have been shown to have multiple embryonic origins, depending on their maturation location: hence, facial and thymic MCs derive from the neural crest (NC), whereas gut, lung, and liver MCs are mesoderm-derived (*Asahina et al., 2011*; *Etchevers et al., 2001*; *Que et al., 2008*; *Wilm et al., 2005*). In the skin, although an MC subset has been recently reported to originate from myeloid progenitors, the major source(s) of MCs remain(s) to be unraveled (*Yamazaki et al., 2017*).

Boundary cap (BC) cells form transient aggregates during embryogenesis at the central/peripheral nervous system (CNS/PNS) interface, at the levels of dorsal entry and ventral exit points of all cranial and spinal nerves. Most BC cells express the genes *Egr2* (also known as *Krox20*) and/or *Prss56* that have been used as molecular markers in functional and tracing studies (*Coulpier et al., 2009*; *Niederländer and Lumsden, 1996*). Fate mapping experiments using a murine *Prss56^Cre* allele have recently revealed that Prss56-positive BC cell derivatives migrate between embryonic day (E) 11.5 and E13.5 along nerve roots into dorsal root ganglia (DRGs), where they give rise to Schwann cells (SCs) and a subpopulation of sensory neurons, and along spinal nerves to reach the skin, where they differentiate into SCs, terminal glia, and melanocytes (*Gresset et al., 2015*; *Radomska et al., 2019*). Tracing experiments performed with an *Egr2^Cre* allele led to similar conclusions concerning nerve roots and DRGs (*Maro et al., 2004*). However, in this case, peripheral migration was not explored. Here, we show that Egr2-positive BC cell derivatives migrate, during the same time period, along spinal nerves to reach the skin. In the skin, most of them detach from nerves, integrate the vascular plexus, and provide the major source of MCs to the adult skin vasculature.

## Results

### Derivatives of Egr2-positive BC cells join the skin vascular plexus after migration along the nerves

To investigate the contribution of Egr2-positive BC cells to the developing trunk skin, we crossed mice carrying *Egr2^Cre* and *Rosa26R^Tom* alleles and traced derivatives, on the basis of tdTomato (Tom) expression in *Egr2^Cre/+*,*Rosa26R^Tom* embryos. As previously reported, we found numerous Tom-positive cells in the dorsal and ventral roots at E11.5 (*Figure 1A*), as well as in the DRGs at E12.5 (*Figure 1B*). In addition, traced cells were observed in the proximal spinal nerve segments at E11.5 (*Figure 1A and D*), in the intermediate segments of dorsal and ventral rami at E12.5 (*Figure 1B and E*) and E13.5

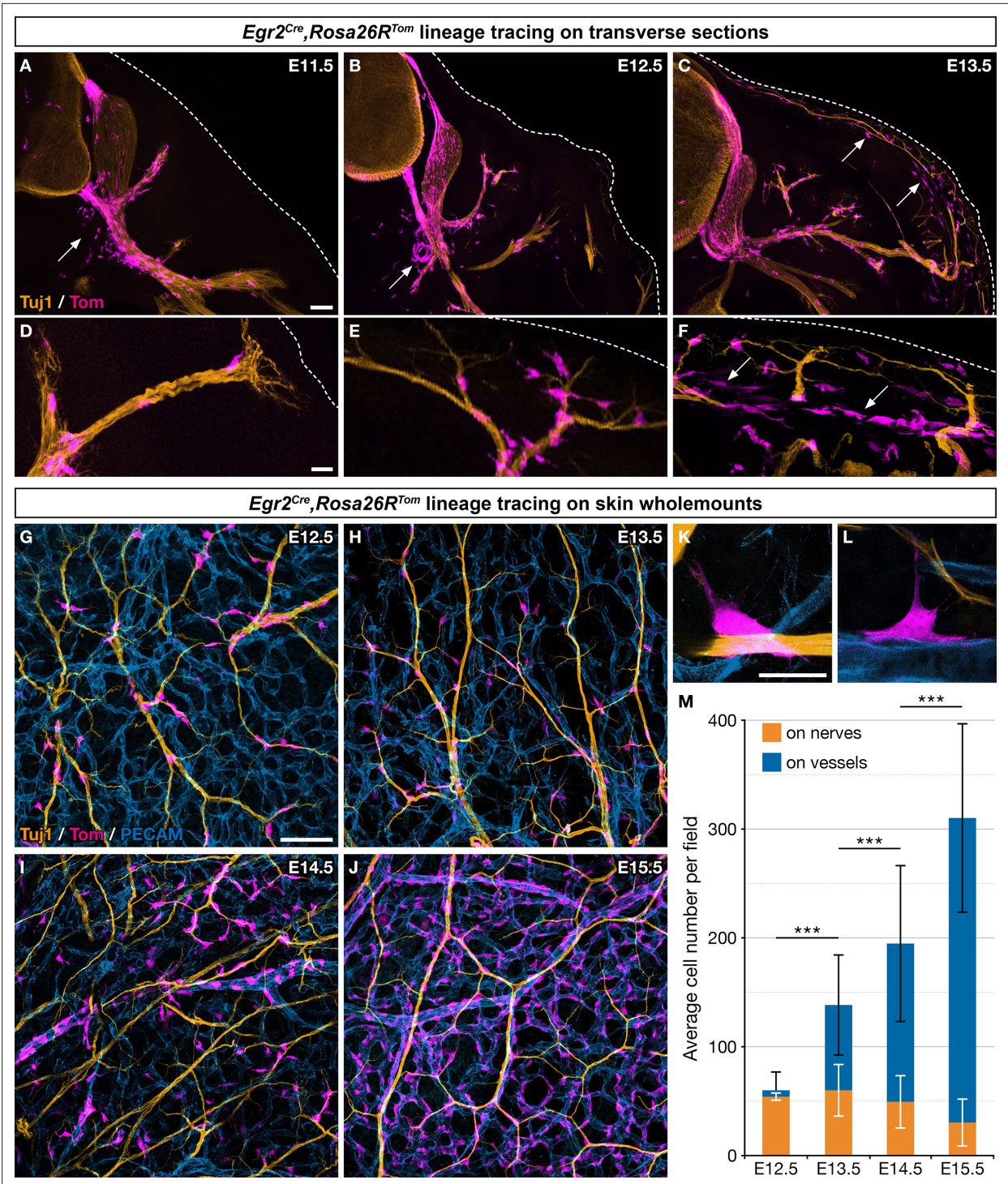

**Figure 1.** Boundary cap (BC) cell derivatives migrate along the nerves and incorporate into the vascular plexus. (**A–F**) Trunk transverse sections from *Egr2^{Cre/+},Rosa26R^{Tom}* embryos at the indicated stages, and labeled with antibodies against Tomato (magenta) and Tuj1 (orange). Cells detached from the nerves (arrows) appear first in the vicinity of the ventral roots (**A,B**) and later close to the skin, as indicated by the dotted line (**C,F**). (**G–J**) Whole-mount dorsal skin from *Egr2^{Cre/+},Rosa26R^{Tom}* embryos at the indicated stages, labeled with antibodies against Tomato (magenta), Tuj1 (orange), and PECAM (blue). (**K,L**) Higher magnifications showing cells in contact with both nerve and vessel at embryonic day (E) 13.5. (**M**) Quantification of the number of labeled cells associated with nerve or vessel per field at the indicated stages (n=3 embryos per stage). Statistical analyses of the "on nerve"/"on vessels" ratio between time points were carried out using a Mann-Whitney U test. Scale bars, 100 µm (**A–C,G–J**) and 20 µm (**D–F,K–L**). Error bars, one standard deviation (**M**). ***=p < 0.001.

*Figure 1 continued on next page*

*Figure 1 continued*

The online version of this article includes the following figure supplement(s) for figure 1:

**Figure supplement 1.** Traced cells, apart from boundary cap (BC) cells themselves, do not express *Egr2* at embryonic day (E) 12.5 and behave differently from Prss56-positive BC cell derivatives.

(*Figure 1C*), and finally at dorsal and ventral skin nerve terminals from E13.5 (*Figure 1C and F*). This dynamic time-course of proximo-distal distributions strongly suggests a migration of derivatives of Egr2-positive BC cells along nerves. We also noticed that some of the traced cells were not in contact with nerves, near the ventral roots at E11.5–12.5 (*Figure 1A and B*, arrows) and later in the skin (*Figure 1C and F*, arrows). To ensure that all traced cells originate from Krox-20-positive BC cells, i.e., that Tom expression does not result from de novo activation of *Egr2* in other cells populations, we searched for *Egr2* expression at E12.5 by in situ hybridization on whole embryos (*Figure 1—figure supplement 1A*), RT-PCR analysis of total RNA extracted skin (*Figure 1—figure supplement 1B*), and RNAscope on whole embryos (*Figure 1—figure supplement 1C–J*). These experiments did not reveal the presence of *Egr2* mRNA apart from BC cells. Furthermore, single-cell transcriptomic analysis of Tom-positive skin cells in E12.5 *Egr2^{Cre/+},Rosa26R^{Tom}* embryos did not reveal any expression of Cre (see below). Together these data indicate the absence of de novo *Egr2* expression in traced cells outside of the BC. In conclusion, these analyses establish that derivatives from *Egr2*-expressing BC cells migrate along spinal nerves into the skin, and that some of them detach from the nerves during their journey.

To further characterize derivatives of Egr2-positive BC cell in the skin, we performed whole-mount immunohistochemistry on embryonic skin between E12.5 and E15.5, staining for axons (Tuj1), blood vessels (PECAM), and traced cells (Tom). Whereas 89.9% (SD 6.5%) of the traced cells were in contact with nerves at E12.5, this proportion rapidly diminished to reach 9.4% (SD 6.3%) at E15.5 (*Figure 1G–J and M*). Conversely, traced cells were found in contact with blood vessels in increasing proportions, from 10.1% (SD 6.4%) at E12.5 up to 90.6% (SD 6.3%) at E15.5 (*Figure 1G–J and L*). Traced cells in contact with capillaries and larger vessels did not express the endothelial marker PECAM, raising the possibility that they might have acquired an MC identity (*Figure 1H–K*). Finally, a few traced cells were observed straddling both nerves and vessels (*Figure 1K and L*). Together these data suggest that most traced cells, after having traveled from the BCs to the skin along nerves, detach from nerves and are recruited within the vascular plexus. Their behavior dramatically differs from that of derivatives of Prss56-positive BC cells (*Figure 1—figure supplement 1K–M and N–P*), which migrate along the same nerves over the same time period, but remain associated with nerves in their vast majority (*Gresset et al., 2015*).

To investigate a possible role of Egr2 in the behavior of derivatives of Egr2-positive cells, we generated and analyzed *Egr2^{Cre/Cre},Rosa26R^{Tom}* mutant embryos, in which the tracing system is combined with the inactivation of *Egr2*. We did not observe any obvious difference in the pattern and number of Tom-positive cells in contact with blood vessels at E14.5, as compared to *Egr2^{Cre/+},Rosa26R^{Tom}* control embryos (*Figure 1—figure supplement 1K–M and Q–S*). These latter data do not support a role of *Egr2* in the migration of BC cell derivatives and in their contribution to the vascular plexus.

## In the skin vasculature, BC cell derivatives express perivascular markers and adopt mural characteristics

To characterize the derivatives of Egr2-positive BC cells in the skin vasculature, we performed immunolabeling analyses using a panel of MC markers. We found that at E14.5, traced cells associated with the vascular network showed characteristic morphologies compatible with pericyte or vSMC identities, as they covered small capillaries or wrapped around larger vessels, both arteries and veins (*Figure 2A–D*). Consistently, these cells expressed several MC markers, such as ABCC9, NG2, PDGFRβ, and smooth muscle actin (SMA) (*Figure 2A–D*). Quantification studies of traced cells in E15.5 skin whole-mounts and in postnatal day 1 (P1) skin sections revealed that they represented respectively 68% (SD 4.6%) and 66% (SD 8.1%) of the NG2-positive MCs (*Figure 2E–G*). Whole-mount analysis of newborn and P30 dorsal skin indicated the persistence of numerous BC cell-derived MCs on capillaries, arteries, and veins (*Figure 2—figure supplement 1A–E*).

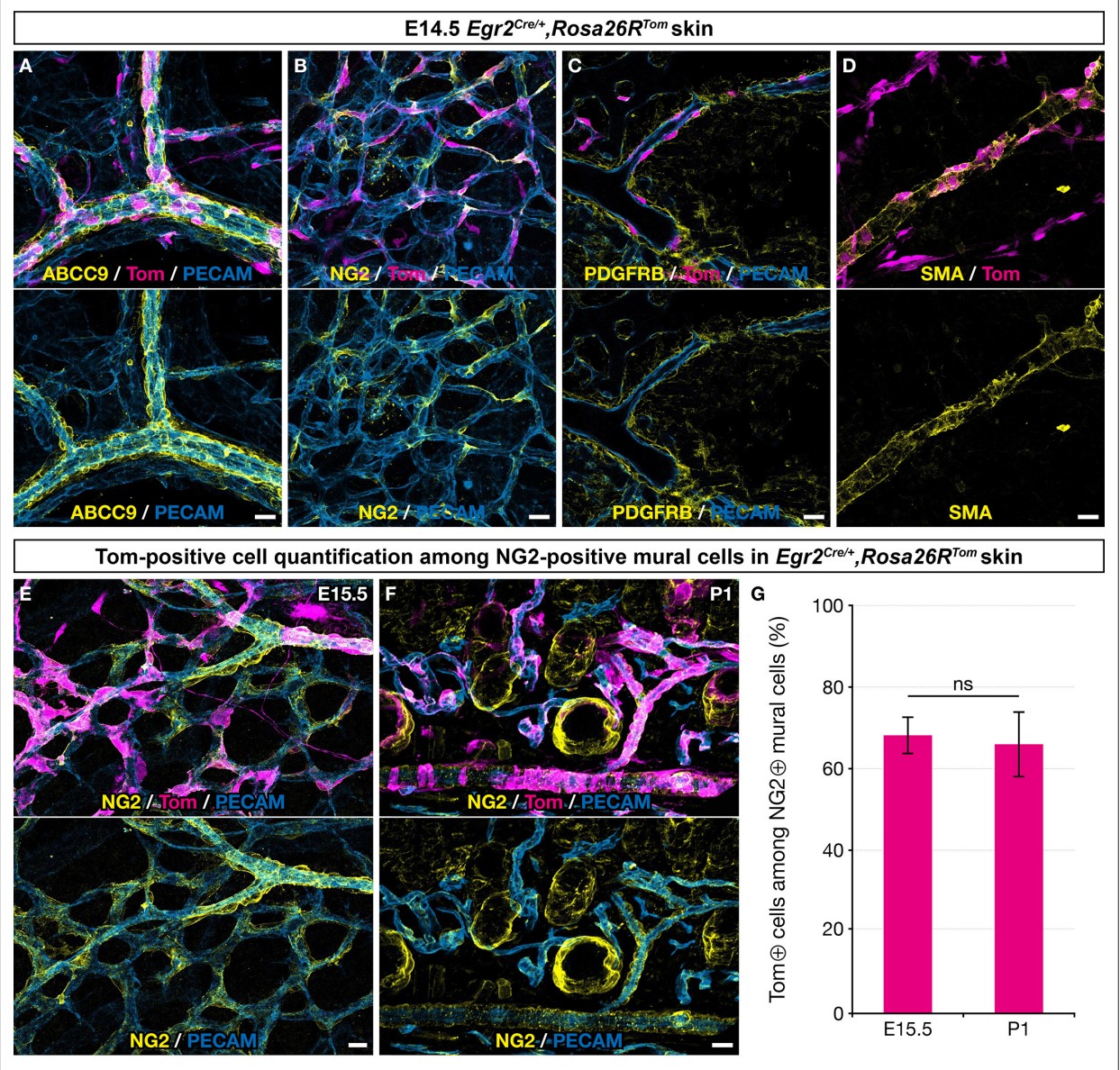

**Figure 2.** Boundary cap (BC) cell derivatives give rise to the majority of skin mural cells (MCs). Whole-mount embryonic dorsal skin at the indicated stages (**A–E**) or transverse section of newborn skin (**F**), labeled with antibodies against Tomato (magenta), PECAM (blue), and ABCC9, NG2, PDGFRβ, or smooth muscle actin (SMA) (yellow). (**A–D**) In the vascular plexus, Tomato-positive cells express MC markers: ABCC9 (**A**), NG2 (**B**), PDGFRβ (**C**), and SMA (**D**). (**E–G**) Tomato-positive cells account for approximately two thirds of NG2-positive MCs in the skin at embryonic day (E) 15.5 (**E,G** ; n=3)and postnatal day (P) 1 (**F,G**; n=3). Tomato labeling is omitted in the lower images (**A–F**). Statistical analysis of the "NG2 and Tomato-positive"/"NG2-positive" ratio between time points was carried out using a Mann-Whitney U test. Scale bars, 20 µm. ns, non-significant (**G**).

The online version of this article includes the following figure supplement(s) for figure 2:

**Figure supplement 1.** Boundary cap (BC) cells give rise to skin mural cells (MCs), including pericytes, arterial and venous vascular smooth muscle cells (VSMCs).

**Figure supplement 2.** Numerous traced mural cells (MCs) are present in serous membranes from various organs.

Overall, these data indicate that, in the skin, derivatives of Egr2-positive BCs detach from nerves, adopt mural characteristics, and constitute the major part of the embryonic and newborn MC component. Furthermore, this BC cell-derived MC population is maintained up to P30 at least.

## A subpopulation of Egr2-expressing BC cells acquires mesenchymal identity before their emigration to the periphery

The immunolabeling analysis presented above revealed that derivatives of *Egr2*-expressing BC cells migrate along peripheral nerves, detach from them, and contribute to the major part of pericytes and vSMCs in the skin. Notably, this population differs from *Prss56*-expressing BC cells, which never give rise to such derivatives. Two main hypotheses can be proposed to explain this process. First, a subpopulation of *Egr2*-expressing BC cells is already specified as MPs before migration to the skin and

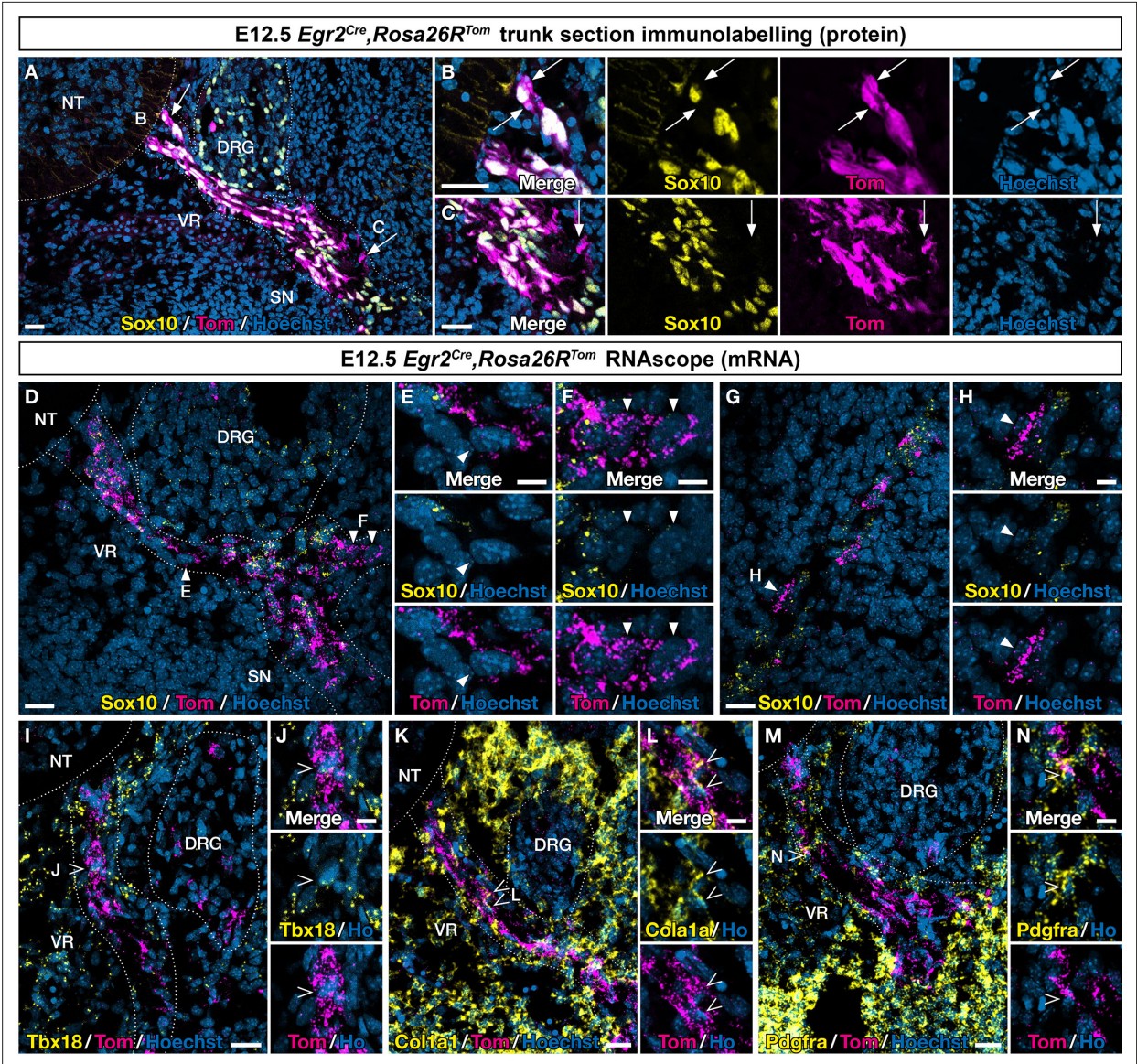

**Figure 3.** An *Egr2*-expressing boundary cap (BC) cell subpopulation displays mesenchymal identity before migrating to the periphery. Trunk transverse sections (**A–F, I–N**) and skin sections (**G,H**) from *Egr2^{Cre/+},Rosa26R^{Tom}* embryos at embryonic day (E) 12.5. Sections were labeled with antibodies (**A–C**) against Tomato (magenta) and Sox10 (yellow), or analyzed by RNAscope in situ hybridization (**D–H**) for *Tomato* (magenta) and *Sox10* (yellow). Detailed analysis revealed that some Tomato-positive cells on ventral roots did not express Sox10 at the protein (arrows) and RNA level (arrowheads). RNAscope in situ hybridization of *Tbx18* (**I,J**), *Col1a1* (**K,L**), and *Pdgfra* (**M,N**) expression in ventral roots from E12.5 embryos. Note that some tomato-positive cells on ventral roots express these markers (empty arrowheads). Scale bars, 25 µm (**A,D,G,I,K,M**) or 10 µm (**B,C,E,F,H,J,L,N**).

their derivatives detach from nerves and differentiate into MCs once reaching their target. Second, *Egr2*-expressing BC cell-derived glial progenitors migrate along nerves to the skin and undergo a glial-to-vascular transition that allows them to detach from nerves and mature into MCs. In both scenarios, nerve detachment would likely be dictated by the microenvironment through the influence of locally secreted factors.

To further explore the identity of *Egr2*-expressing BC cells and their derivatives, we first analyzed E12.5 ventral nerve roots to assess whether all Tom-positive cells were expressing the glial marker Sox10, at the protein or mRNA levels (*Figure 3A–C*). We observed a majority of cells positive for both tomato and the Sox10, protein, but also some nerve-attached Tom-positive/Sox10-negative cells (*Figure 3A–C*). Similar results were obtained by RNAscope, combining *Tomato* and *Sox10* probes (*Figure 3D–F*). However, since *Sox10* activation occurs progressively in early-stage glial progenitors, we could not exclude that Tom-positive/Sox10-negative cells could correspond to early glial progenitors that have not yet activated this marker. This led us to repeat this analysis on skin nerve endings at E12.5 (*Figure 3G and H*). Similarly, we found among nerve-attached Tom-positive cells a majority of Sox10-positive cells, but also some Sox10-negative cells. An additional argument in favor of an early commitment of some ventral root cells toward a Sox-10-negative fate came from the analysis of the dorsal root (data not shown). In this case, we never observed Tom-positive/Sox10-negative cells. This is consistent with the fact that Tom-positive cells do not detach from the dorsal root.

To determine whether the presence of Sox10-negative cells would be accompanied by the expression of MP markers on ventral nerve roots, we used RNAscope to assess the expression of *Tbx18*, *Col1a1*, and *Pdgfra*. As triple RNA labeling was technically challenging, we could only perform double labeling, with Tomato and one marker at a time. We found that each of these three markers was co-expressed with Tomato in some cells (*Figure 3I–N*).

In conclusion, in the ventral root, these findings support the existence of cells of mesenchymal identity derived from *Egr2*-expressing BC cells. These derivatives further migrate along the nerves to reach the skin, then detach and differentiate into MCs. In the dorsal root, such cells would not exist, consistent with the absence of cell detachment from the nerve.

## Ventral BCs are comprised of cells from distinct embryonic origins

Although fate mapping studies performed in chick and mice point to the NC as the population at the origin of dorsal root BC cells, the situation appears more complex in the case of the ventral root population (*George et al., 2007*; *Kucenas et al., 2008*; *Yaneza et al., 2002*). Moreover, only cranial NC was shown to give rise to MCs in the head vasculature and similar potential for thoracic NC was never reported, despite efforts from many teams. To identify the embryonic origin of ventral root the *Egr2*-expressing BC cells that give rise to skin MCs, we performed genetic fate mapping analyses using Cre drivers targeting the NC (*Wnt1^Cre^, Sox10^Cre-Ert2^*) and the ventral neural tube (*Olig2^Cre^*), which are in the vicinity of BCs, in combination with *Rosa26R^Tom^* or *Rosa26R^YFP^* Cre-inducible reporters. In the case of *Sox10^Cre-Ert2^*, tamoxifen induction was performed at E9.5, which corresponds to the period of NC delamination. Embryos carrying the different Cre drivers were collected between E12.5 and E14.5 and analyzed by immunohistochemistry. In all cases, we were not able to identify any traced MCs, either near ventral roots or in the skin (*Figure 4A–F*), in accordance with previous studies that have shown that the NC and neural tube do not contribute to the trunk vasculature (*Etchevers et al., 2019*). We noticed that a subpopulation of cells derived from BC cells co-express *Tbx18* in the ventral root (*Figure 3G–L*). Furthermore, at E9.5, we detected *Tbx18* expression in the vicinity of the neural tube, in a layer of MPs that gives rise to fibroblasts in the meninges (*Figure 4—figure supplement 1*), consistent with a previous report (*DeSisto et al., 2020*). We therefore performed a genetic fate mapping of Tbx18 lineage using a *Tbx18* Cre driver line (*Tbx^Cre-ERT2^*). Tamoxifen was administrated to pregnant females at E9.5 and embryos were analyzed at E11.5 and E14.5. At E11.5, we observed Tom-positive/Sox10-negative cells lining the ventral root and the spinal nerve with some of them closely attached to nerves (*Figure 4G–L*). At E14.5, we observed numerous Tom-positive cells along blood vessels co-expressing NG2 or/and SMA (*Figure 4—figure supplement 2A–F, J–L*). None of them express Sox10 (*Figure 4—figure supplement 2G–I, M–O*).

Together our observations suggest a dual embryonic origin of ventral root BCs expressing Egr2: part of this population originate from the NC, while another may originate from the pial cells. These

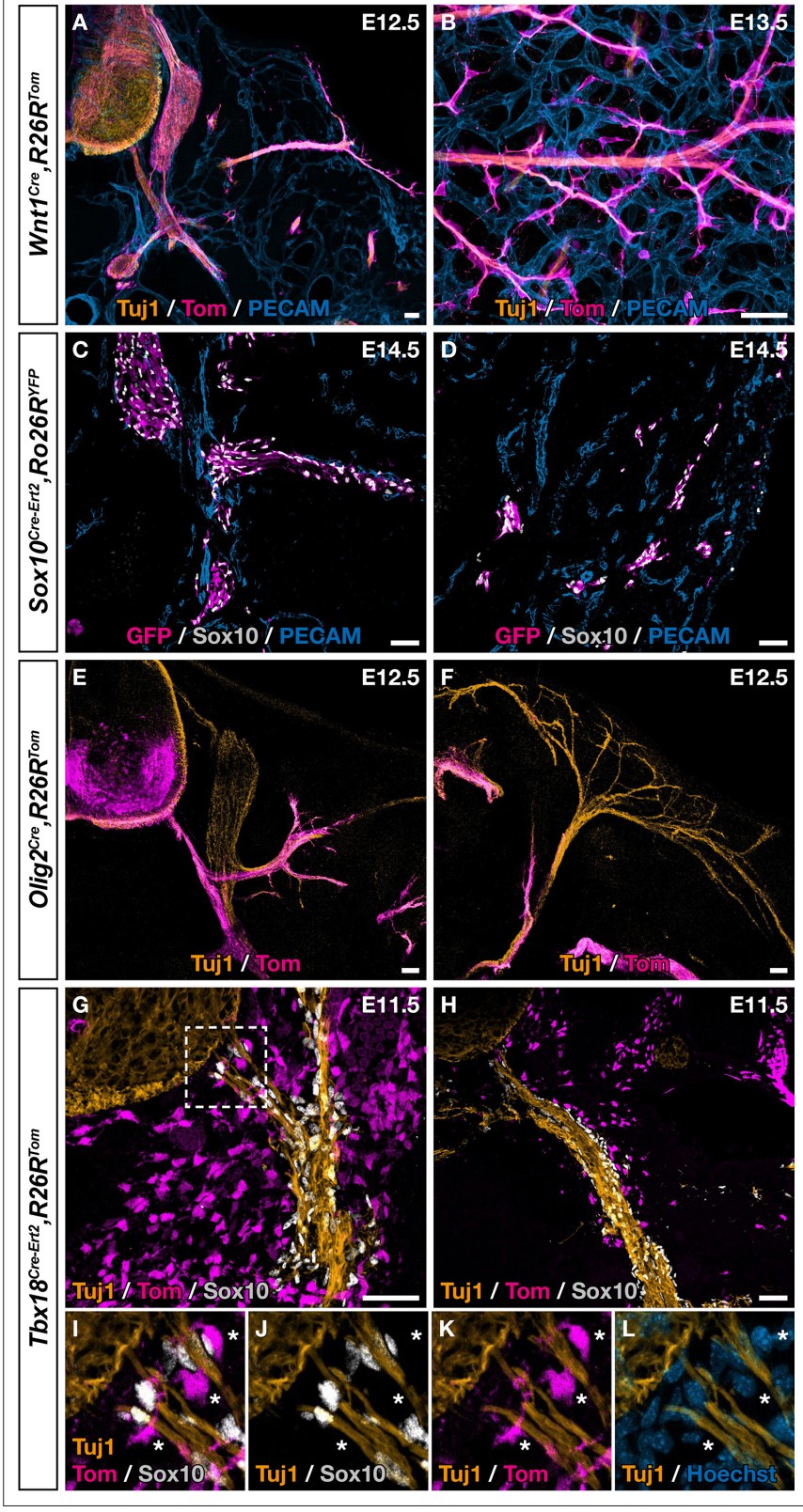

**Figure 4.** Ventral boundary caps (BCs) have a dual embryonic origin. Trunk transverse sections (**A,C,E–L**), dorsal skin whole-mount (**B**), and skin section (**D**) at the indicated embryonic stages. Genetic fate mapping of neural crest (NC) using $Wnt1^{Cre/+},Rosa26R^{Tom}$ (**A,B**) and $Sox10^{Cre-Ert2/+},Rosa26R^{Tom}$ (**C,D**) embryos and of neuroepithelial cells and their derivatives using $Olig2^{Cre/+},Rosa26R^{Tom}$ embryos (**E,F**) did not reveal any tomato-positive cells attached to the

*Figure 4 continued on next page*

*Figure 4 continued*

vascular plexus, neither close to ventral roots nor to nerves in the skin. (**G,L**) Genetic fate mapping of developing pia matter using *Tbx18^Cre-Ert2/+^,Rosa26R^Tom^* embryos revealed the presence of numerous tomato-positive, Sox10-negative cells in close contact with the nerve (asterisks, **I–L**). Tamoxifen was delivered to pregnant females at embryonic day (E) 9.5 (**C,D,G–L**). Scale bars, 50 μm.

The online version of this article includes the following figure supplement(s) for figure 4:

**Figure supplement 1.** *Tbx18* is expressed in cells around the developing neural tube.

**Figure supplement 2.** *Tbx18*-expressing cells contribute to mural cells (MCs) in the skin and around the neural tube.

---

latter population, once in contact with the ventral root, activate *Egr2*, do not express *Sox10,* and then undergo migration along the nerves to the skin and mature into perivascular derivatives.

## BCs are comprised of cells with distinct glial and mesenchymal signatures

The above analyses support the existence of a subpopulation of *Egr2*-expressing BC cells already specified as MPs, before migrating into the skin. To further characterize the molecular identity and the fate of their derivatives, we performed a single-cell transcriptomic analysis (single-cell RNA sequencing [scRNA-seq]) of traced Egr2-positive cells. As the quantification of traced cells in contact with nerves or blood vessels indicated that detachment was initiated at E12.5 (*Figure 1M*), we focused on this stage, on Tom-positive cells isolated by fluorescence-activated cell sorting (FACS) from dissociated skin. Using a customized pipeline (Materials and methods), we were able to analyze 2209 single-cell transcriptomes with a mean number of expressed genes per cell of 5424 (SD 1491). As indicated above, to ensure that *tomato* expression was not due to ectopic *Cre* expression, we inserted the *Cre* sequence in our scRNA-seq analysis. We did not observe any *Cre* expression. Using the Seurat software (*Stuart et al., 2019*), we identified 10-cell clusters, based on their expression patterns (*Figure 5A*). To determine their molecular identities, we overlaid plotted cells with levels of expression of well-characterized marker genes (*Figure 5B–D*). This approach allowed us to regroup the initial clusters into three main cell type-defined supra-clusters: an SCP supra-cluster, a mesenchymal progenitor (MP) supra-cluster, and a smaller, isolated MC cluster (*Figure 5A*). The SCP supra-cluster contained 1031 cells (47%) that expressed high levels of SCP markers, such as *Sox10*, *Erbb3*, *Fabp7*, *Plp1*, *Ngfr,* and *Cdh19* (*Figure 5B*). The MP supra-cluster contained 1056 cells (48%) that expressed a combination of mesenchymal markers, such as *Pdgfra*, *Pdgfrb*, *Dlk1*, *Tbx18*, *Igf1,* and *Col1a1* (*Figure 5C*). Interestingly, whereas most traced cells (90%) were in contact with nerves at E12.5 (*Figure 1G and M*), 54% of them showed either low or undetectable levels of expression of glial markers, such as *Sox10*, and significant expression of mesenchymal markers (*Figure 4B–C*). This finding is in line with our previous observation that a subpopulation of Tom-positive cells lining the nerves do not express *Sox10* (*Figure 3A–D'*). Finally, the isolated MC cluster contained 122 cells (6%) that expressed high levels of *Pdgfrβ*, *Abcc9*, *Acta2*, *Cspg4,* and *Rgs5* (*Figure 5C and D*), which collectively constitute a signature for pericytes and vSMCs. This percentage is similar to the proportion of traced cells attached to blood vessels at E12.5 (*Figure 1M*). Therefore, the latter cells may have achieved their maturation into MCs.

Further analyses using additional markers revealed a significant heterogeneity within the supra-clusters, in line with the identification of the 12 original clusters (*Figure 5—figure supplement 1A–C*). Hence, the SCP supra-cluster regroups four clusters (SCP1–4) that differ in the expression of SCP markers, such as *Gap43*, *Pou3f1,* and *Pax3* (*Figure 5—figure supplement 1B*). Similarly, the MP supra-cluster regroups five clusters: four of them (MP1–4) differentially express some MC progenitor genes, including *Tbx18*, *Pax9*, *Kcnj8, or Acta2* (*Figure 5C and D* and *Figure 5—figure supplement 1C*), whereas MP5 displays a transcriptomic signature of chondroprogenitors with markers such as *Sox9*, *Dlx5*, *Col2a1,* and *Fgfr2* (*Figure 5—figure supplement 1D*). These results suggest that in addition to mural progenitors, Egr2-positive BC cells may contribute to chondroprogenitors in the trunk.

In conclusion, the single-cell RNA analysis corroborates and extends the immunolabeling data presented above, and establish that Egr2-positive BC cells consist of two major subpopulations: a first one which evolves into SCP and is likely to derive from the NC, and a second one, which gives rise

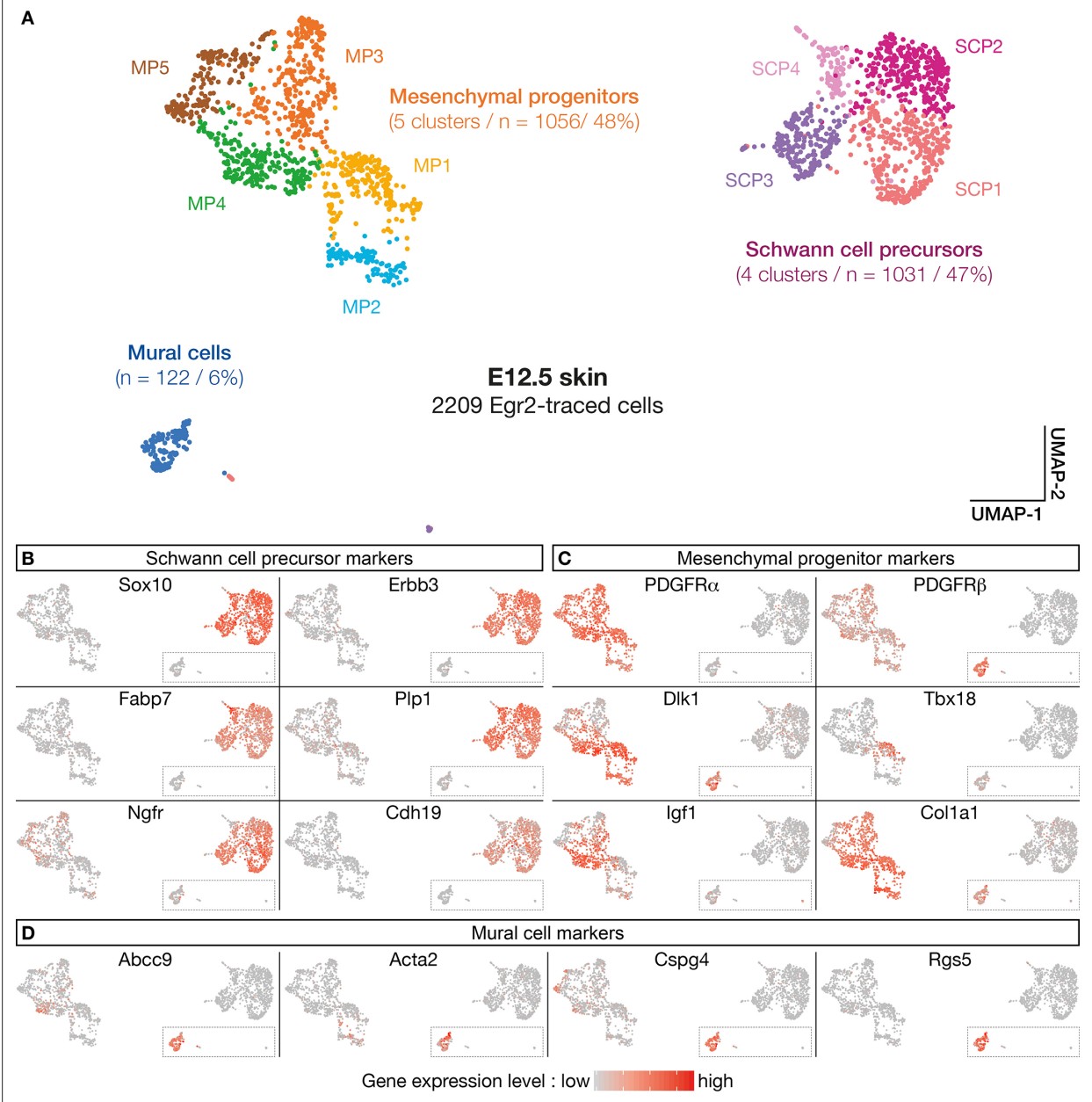

**Figure 5.** Numerous molecular features reveal diversity in cell identity in embryonic day (E) 12.5 boundary cap (BC) derivatives. (**A–D**) 10 clusters can be delineated on the basis of single-cell RNA sequencing (scRNA-seq) experiments performed on traced cells from E12.5 dorsal skin. They can be organized in three supra-clusters based on their molecular signatures: Schwann cell precursors, mesenchymal progenitors, and mural cells. In (**B–D**), cells are color-coded, with color intensities (from gray to red) reflecting the relative levels of expression of each gene.

The online version of this article includes the following figure supplement(s) for figure 5:

**Figure supplement 1.** Heterogeneity of the SPC and mesenchymal progenitor (MP) supra-clusters.

to derivatives with an MC identity and may originate from the mesenchymal precursors of meningeal fibroblasts.

## Derivatives of Egr2-positive BC cells contribute to the adult vasculature of several serous membranes

As indicated above, in *Egr2^{Cre/+},Rosa26R^{Tom}* embryos, we observed Tom-positive cells without nerve contact, near the ventral roots at E11.5 and at the level of the developing pulmonary artery at E12.5 (*Figure 1A and B*, arrows). These findings suggested a broader implication of BC cells in

trunk vasculature development, and led us to search more systematically for Tom-positive MCs. We performed a detailed study of two adult $Egr2^{Cre/+},Rosa26R^{Tom}$ mice and analyzed various organs (thyroid gland, lungs, heart, esophagus, stomach, intestines, liver, spleen, and kidneys) and serous membranes (pre-tracheal fasciae, pericardium, and peritoneum). Whereas we did not observe traced MCs in these organs, we found numerous Tom-positive MCs in the serous membranes that ensheath some of them, as revealed by co-labeling with PECAM of NG2 (*Figure 2—figure supplement 2A–C*). As described in the skin, Tom-positive MCs do not form a continuum and are separated by non-traced MCs, suggesting at least another embryonic origin for MCs in serous membranes. While these latter data demonstrate that serous membrane MCs derive from *Egr2*-expressing cells, in the absence of additional data on earlier *Egr2* expression, it remains to be established whether they actually originate from BC cells. Nevertheless, the latter hypothesis is attractive, considering the precedent of the skin and the presence of such cells at the level of the pulmonary artery at E12.5. In this situation, BC cells would provide a widespread contribution to the adult vasculature.

## BC cell ablation results in a depletion in skin MCs

As about two thirds of newborn skin MCs originate from *Egr2*-expressing BC cells (*Figure 3F and G*), we questioned the functional importance of this population in blood vessel development. To address this issue, we performed a genetic ablation of *Egr2*-expressing BC cells. In $Egr2^{Cre/GFP(DT)},Rosa26R^{Tom}$ embryos, Cre-inducible diphtheria toxin (DT) expression is restricted to *Egr2*-expressing cells, including BC cells, leading to their rapid elimination (*Vermeren et al., 2003*). We noticed that genetic ablation of *Egr2*-expressing cells induced embryonic lethality at around E15.5, as we were unable to obtain mutant embryos beyond that stage, and we often observed sites of necrosis in the uterus when collecting embryos at E15.5. At E15.5, among the 60 collected embryos, issued from $Egr2^{Cre/+},Rosa26R^{Tom} \times Egr2^{GFP(DT)}$ breeding, 10 were $Egr2^{Cre/GFP(DT)},Rosa26R^{Tom}$ based on PCR genotyping. We analyzed these embryos for tomato fluorescence using a stereomicroscope. Unexpectedly, we observed traced cells in the skin in six of these embryos, suggesting that BC cell ablation was not complete. The four others displayed a complete loss of traced cells and were used for further analyses of the vascular network. Skin whole-mount immunohistochemistry for blood vessels (PECAM) and MCs (NG2) were performed on these latter mutants and three littermate controls ($Egr2^{Cre/+},Rosa26R^{Tom}$) embryos (*Figure 6A–F*). The skin vascular network appeared grossly similar in both types of embryos, with a dense network of arterioles, venules, and capillaries (*Figure 6A–C, E, and G*). However, when we compared the fraction of the vascular area covered by NG2-positive MCs (*Figure 6D, F, and H*), we observed a significant reduction in the mutants (6.5%, SD 4.3%), as compared to the controls (17.9%, SD 2.0%). We also explored whether *Egr2*-expressing cell ablation impacted nerve development and performed skin whole-mount immunohistochemistry for nerves (Tuj1) and SCPs (Sox10) at E15.5 in two controls and two mutants with complete loss of traced cells. There was no significant difference between littermate controls ($Egr2^{Cre/+},Rosa26R^{Tom}$) and mutants in terms of Tuj1-positive area fraction and Sox10-positive cell count (*Figure 6—figure supplement 1A–C*).

In conclusion, we find that the targeted ablation of *Egr2*-expressing cells does not appear to impact nerve development, but leads to a significant depletion of NG2-positive MCs in developing capillaries in the skin. This indicates that MCs from myeloid origin cannot fully compensate this defect. It remains to be determined whether this vascular phenotype is at the origin of embryonic lethality, as ablation targets all Egr2-positive cells and is therefore not limited to BC cells.

## Discussion

This study uncovers a dual embryonic origin for MCs in mouse skin: while approximately one third of this population derives from myeloid precursors (*Yamazaki et al., 2017*), about two thirds originate from BCs. Hence, we identified a specific subpopulation derived from *Egr2*-expressing, ventral BCs that migrate along nerves into the skin, then detach and give rise to pericytes and vSMCs. The remaining BC-derived cells follow the same path, but remain attached to nerves and give birth to SCPs. Whereas these latter cells express the typical NC lineage marker Sox10, our observations suggest that the former population giving rise to MCs do not express this marker and originates from the developing pia matter. In accordance with our tracing studies, genetic ablation of *Egr2*-expressing BC cells results in a depletion of the skin MC compartment. Finally, we found that numerous MCs in serous

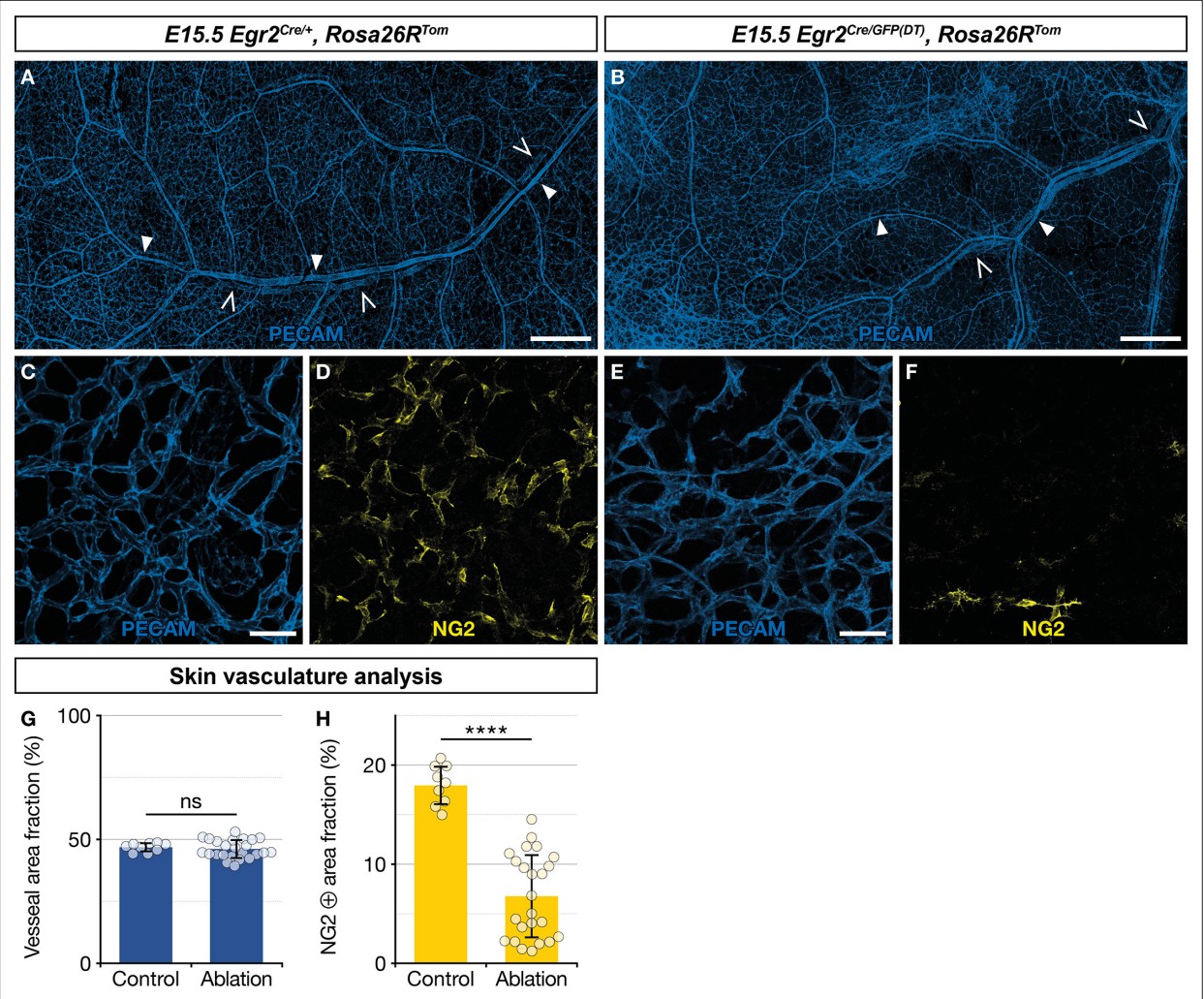

**Figure 6.** Ablation of *Egr2*-expressing boundary caps (BCs) impacts mural cell (MC) development in the skin. Whole-mount embryonic dorsal skin at embryonic day (E) 15.5, labeled with antibodies against PECAM (blue) and NG2 (yellow). Compared with littermate *Egr2^Cre/+^,Rosa26R^Tom^* control mice (**A**), vascular network development appeared normal at low magnification in mutant mice (**B**), in which BC cells and their derivatives were eliminated (arterioles and venules are indicated by full and empty arrowheads, respectively). Capillaries from mutant mice skin (**E,F**) showed a significant decrease in NG2-positive MCs as compared to controls (**C,D**). Vessel area fractions were similar in controls and mutants (**G**), but the NG2-positive area fraction was significantly decreased in the mutants (**H**), indicating a depletion of MCs. Statistical analyses of the PECAM and NG2-positive surface areas between controls (n=3) and mutants (n=4) were carried out using a Mann-Whitney U test. Scale bars, 500 μm (**A,B**), 50 μm (**C–F**). Error bars, one standard deviation (**G,H**). ****=p < 0.0001. ns, non-significant.

The online version of this article includes the following figure supplement(s) for figure 6:

**Figure supplement 1.** Ablation of *Egr2*-expressing boundary caps (BCs) does not affect normal skin innervation.

membranes protecting the trachea, heart, and digestive tract are derived from *Egr2*-expressing cells, suggesting a broader implication of BCs in the development of the systemic vasculature.

Our observations highlight a novel role for BCs in peripheral vascular development and further expand peripheral glia recently reported broad potential (*Adameyko et al., 2009*; *Dyachuk et al., 2014*; *Espinosa-Medina et al., 2014*; *Furlan et al., 2017*; *Kaucka et al., 2016*; *Uesaka et al., 2015*; *Xie et al., 2019*). While virtually all BC cells at the dorsal root express *Sox10* and give rise to either SCP or DRG sensory neurons, the situation appears more complex in the ventral roots. We discovered that ventral roots encompass two populations of *Egr2*-expressing BCs with distinct molecular and functional profiles. In addition to the Sox10-positive BCs at the origin of SCPs that migrate along nerve roots and spinal nerves, we identified a population of Sox10-negative BC-derived cells that express mesenchymal markers such as Tbx18, Col1a1, and Pdgfra. Our scRNA-seq analysis of traced

cells extracted from E12.5 skin, at a time when the majority of BC-derived cells are still in contact with nerves, further supports the presence of nerve-attached BC derivatives with mesenchymal signature. Interestingly, at the same time, myeloid precursors are recruited by the primitive vascular plexus to form MCs (*Yamazaki et al., 2017*). In both systems, traced cells along blood vessels do not form continuous blocks, but are interrupted by blocks of non-traced cells. This suggests the existence of an attraction mechanism operated by the developing vascular plexus and acting on neighboring cell populations of at least two distinct origins, which have the potential to mature into MCs. In this scenario, nerves would act as a track for the migration of one of these populations from the ventral BCs to the skin and probably some deeper structures in the developing embryo. Since this migration of BC derivatives was observed along all spinal nerves, this mechanism is likely to provide an important source of MCs to the developing embryonic skin. Notably, *Egr2* inactivation does not appear to have any obvious effect on the migration and differentiation of this population. Nevertheless, more detailed analyses will be required to exclude any later function in MCs.

BCs are cell clusters, defined by their location at the CNS/PNS boundary and which were shown to express specific markers such as *Egr2* and *Prss56* (*Coulpier et al., 2009*). Using genetic tracing with the *Prss56* gene, we have previously established that a population of BC-derived traced cells give rise to SCs, DRG sensory neurons, and, in the skin, to terminal glia and melanocytes (*Gresset et al., 2015*; *Radomska et al., 2019*). Genetic tracing performed with the *Egr2*, also revealed a BC cell population that gives rise to DRG neurons and SCs (*Maro et al., 2004*). In this study, we establish that the ventral BC cell population traced with the *Egr2* marker also gives rise to skin MCs, derivatives that are never observed in the case of *Prss56* tracing. The common derivatives observed in the two tracing studies are likely to reflect an already noticed, partial overlap between Prss56- and Egr2-positive BC subpopulations (*Gresset et al., 2015*). In contrast, the differences in fates observed in the skin points to the existence of a functional heterogeneity within BC cells. The developmental fates of BC cells might be determined by the encounter with environmental cues during their migration or, in contrast, by the existence of blueprints established at or even before the BC cell stage. Our studies support such an early specification in the development of ventral BC cells; genetic tracing of cells expressing *Tbx18* indicate that some of them attach to ventral roots before E11.5, where they do not express *Sox10* and may activate *Egr2*. It is likely that these cells contribute to the *Egr2*-traced subpopulation that migrates along nerves to the skin, detaches and differentiates into MCs.

In accordance, our findings require to revisit the simplistic concept of BCs as transient and homogenous clusters of stem-like cells, characterized by co-expression of a panel of markers including *Egr2* and *Prss56* and initially thought to be entirely derived from the NC. The present data suggest that ventral exit point BCs possess mixed NC and mesenchymal origins. In particular, we show that the *Egr2*-expressing BC cells that give rise to MCs are neither derived from the NC nor from the neural tube. They are likely to originate from *Tbx18*-expressing cells that are also at the origin of developing meningeal fibroblasts (*DeSisto et al., 2020*). Actually, molecular heterogeneity was also observed within NC-derived BCs, as they contain populations expressing either *Egr2* or *Prss56*, or co-expressing both markers (*Gresset et al., 2015* and unpublished results). This heterogeneity in the embryonic origin of BC cells is likely to explain another observation: whereas targeted ablation of NC-derived cells expressing *Egr2*, obtained by combining *Wnt1^{Cre}* and *Egr2^{GFP/DT}* alleles, is effective, but not lethal before birth (*Odelin et al., 2018*), targeted ablation of all *Egr2*-expressing cells does lead to embryonic death at around E15.5 (this study). This may be explained by the fact that the populations ablated in those experiments do not completely overlap.

Our demonstration of a novel source of MCs in the skin is of interest in the context of recent advances in understanding MC lineages. Based on their location, MCs have been divided into pericytes that ensheath capillaries, and vSMCs that cover arterioles, venules, and larger vessels (*Armulik et al., 2011*). However, this simplistic view has been recently challenged by growing evidence supporting a continuum of MC lineages with intermediate types associated to specific functions (*Holm et al., 2018*). For instance, molecular analysis of CNS MCs at the single-cell level revealed the existence of subpopulations specifically involved in processes such as inflammation (*Duan et al., 2018*) or fibrosis (*Göritz et al., 2011*; *Soderblom et al., 2013*). A similar heterogeneity among NG2-positive MCs has recently been reported in the skin, with different subpopulations at specific locations and with differing roles in wound repair (*Goss et al., 2021*). Our study reveals the existence of at least two embryonic origins for both skin pericytes and vSMCs, BCs derivatives and myeloid precursors, which are recruited during

the same period to form MCs. Each population appears to be unable to fully compensate the other upon depletion: we report that targeted ablation of BC cells results in a significant decrease in NG2-positive MCs at E15.5 and, conversely, Yamazaki and colleagues have shown that in *PU.1* (also known as *Spi1*) mutants, depleted of tissue myeloid cells, there is also a loss of MCs at E15.5 (*Yamazaki et al., 2017*). In addition, our data show molecular differences between MP clusters susceptible to give rise to MCs in the peripheral vasculature, with most MP clusters (1–4) sharing molecular markers (*Pdgfrα* and *Col1a1*) with CNS MC subtypes involved in fibrosis. Whether these differences in embryonic origin or marker expression identify distinct functional lineages corresponding to those reported at later stages remains an open question.

## Materials and methods

### Mouse lines, genotyping, and ethical considerations

Mice used in this study were housed in a temperature- and humidity-controlled vivarium on a 12 hr dark-light cycle with free access to food and water. All mouse lines were maintained in a mixed C57BL/6-DBA2 background. We used the following alleles or transgenes that were genotyped as indicated in the original publications: *Egr2^Cre^*(*Voiculescu et al., 2000*), *Prss56^Cre^* (*Gresset et al., 2015*), *Wnt1^cre^* (*Lewis et al., 2013*), Sox10X^Cre-Ert2^ (*Simon et al., 2012*), *Olig2^Cre^* (*Zawadzka et al., 2010*), *Tbx18^Cre-Ert2^* (*Guimarães-Camboa et al., 2017*), *Rosa26R^YFP^* (*Srinivas et al., 2001*), *Rosa26R^Tom^* (*Madisen et al., 2010*), and *Egr2^GFP(DT)^* (*Vermeren et al., 2003*). Day of plug was considered E0.5. Animals were sacrificed by decapitation (newborn) or cervical dislocation (adult) unless indicated otherwise. All animal manipulations were performed according to French and European Union regulations. According to these regulations, no ethics committee approval was required for this study which only used mouse embryos and newborns.

### In situ hybridization and immunofluorescence

In situ hybridization on embryo sections was performed as previously described (*Gresset et al., 2015*). Briefly, samples were fixed overnight (o/n) in 4% paraformaldehyde (PFA; Electron Microscopy Science) in 0.1 M phosphate buffer (PBS) before being serially sectioned (150 μm) and processed for in situ hybridization. Embryonic immunohistochemical analysis was performed either on 50 μm transverse cryosections or embryonic skin whole-mounts, both performed as previously described (*Gresset et al., 2015*; *Radomska et al., 2019*). Sections and embryonic skin whole-mounts were stored at –20°C in 0.1 M PBS with 30% glycerol and 30% ethylene glycol. Briefly, for whole-mount immunolabeling, dorsal skins of E12.5–15.5 embryos were dissected after o/n fixation in 4% PFA. Samples were blocked o/n in 4% bovine serum albumin (BSA, Sigma-Aldrich) in PBS containing 0.3% Triton X-100 (PBST, Sigma-Aldrich), then incubated for 3 days with the primary antibody/BSA/PBST solution at 4°C. After rinsing, secondary antibodies were applied o/n at room temperature. Samples were then washed and flat-mounted in Fluoromount-G (Southern Biotech). Antibodies are described below. Nuclei were counterstained with Hoechst (Life Technologies). Whole-mount immunostaining and clarification of newborn skin was performed using the iDISCO+ method (*Renier et al., 2016*). Z-stacks were acquired using Leica TCS SP5 and TCS SP8 laser-scanning confocal microscopes and assembled in ImageJ.

### RNAscope in situ hybridization

We used an RNAscope Multiplex Fluorescent Reagent Kit V2 and Target Probes (ACD Biotechne) for mouse high-resolution in situ hybridization detection of *Collagen1a1, Egr2, Pdgfra, Sox10, Tbx18*, and *Tom* mRNAs (ACD Biotechne). The experiment was performed according to the manufacturer's instructions. In an RNase-free environment, 14-μm-thick slices of fixed WT embryos at E9.5 and E10.5, and *Egr2^Cre/+^,Rosa26R^Tom^* embryos at E12.5, were sectioned and mounted on a glass slides. Sections were baked in a 60°C dry oven for 30 min, post-fixed in 4% PFA for 2 hr, then dehydrated in 100% ethanol. Sections were incubated for 10 min with a 30% $H_2O_2$ solution before a 5 min incubation in a 1× target retrieval solution at 90–95°C. Sections were then rinsed with distilled water, followed by 100% ethanol. Subsequently, sections were incubated with the following solutions in a HybEZ humidified oven at 40°C with rinsing steps in between: protease III, 30 min; target probes, 2 hr; amplification

(Amp), 30 min; Amp 2, 30 min; Amp 3, 15 min; HRP-OPAL Dye (HRP-C1), 60 min; HRP-C2, 60 min; HRP-C3, 60 min. All reagents were from the RNAscope Multiplex Fluorescent Reagent kit.

## Antibodies

For immunofluorescence, the following primary antibodies were used: rabbit anti-Tomato (1:500, Rockland #600-401-379), goat anti-Tomato (1:500, Sicgen, #AB0040-200), mouse biotinylated anti-Tuj1 (1:800, R&D Systems, #BAM1195), rabbit anti-Tuj1 (1:1000, BioLegend, #802001), rat anti-PECAM (1:100, BD Pharmingen, #553370), goat anti-PECAM (1:1000, R&D Systems, #AF3628), goat anti-Sox10 (1:200, SantaCruz Biotechnology, #sc-17342), rabbit anti-NG2 (1:200, Merck, #AB5320), mouse anti-SMA-Cy3 (1:400, Merck, #C6198), goat anti-PDGFRβ (1:500, R&D Systems, #AF1042), rabbit anti-Abcc9 (1/100, ThermoFisher Scientific, #PA5-52413), and rabbit anti-Desmin (Abcam, #ab15200). Fluorophore-conjugated secondary antibodies were from Jackson ImmunoResearch.

## Cell quantification

Quantification of Tomato-positive cells was performed on whole-mount preparations of embryonic skin from three $Egr2^{Cre/+},Rosa26R^{Tom}$ embryos from each stage (E12.5, E13.5, E14.5, and E15.5), labeled for Tomato, Tuj1, and PECAM. For each embryo, five z-stacks of different and non-overlapping fields of view were selected randomly and acquired using a Leica TCS SP5 laser-scanning confocal microscope. The scanned surface area per field corresponds to 0.38 µm$^2$. All Tomato-positive cells were counted in each stack on the ImageJ software, and among them cells in contact with nerves were counted to calculate the proportion of cells 'on nerve'. Cells not in contact with nerves were on the vascular plexus. Statistical analyses of the 'on nerve'/'on vessels' ratio between time points were carried out using a Mann-Whitney U test.

Quantifications of Tomato-positive cells among NG2-positive cells in $Egr2^{Cre/+},Rosa26R^{Tom}$ mice were performed on whole-mount skin preparations at E15.5 and transverse skin sections at P1, labeled for Tomato, PECAM, and NG2. At E15.5, four non-overlapping z-stacks of a scanned surface area of 0.10 µm$^2$ each were analyzed. At P1, five sections were analyzed, each with two non-overlapping z-stacks of a scanned surface area of 0.10 µm$^2$. All NG2-positive cells and among them Tomato-positive cells were counted in each stack on the ImageJ software. Statistical analysis of the 'NG2 and Tomato-positive'/'NG2-positive' ratio between time points was carried out using a Mann-Whitney U test.

Quantifications of Sox10-positive cells in mutant $Egr2^{Cre/GFP(DT)},Rosa26R^{Tom}$ (n=2) and control (n=2) mice were performed on whole-mount skin preparations at E15.5, labeled for Sox10 and Tuj1. For each embryo, three (controls) or four (mutants) non-overlapping z-stacks of a scanned surface area of 0.20 µm$^2$ were acquired using a Zeiss LSM 900 laser-scanning confocal microscope. All Sox10-positive cells were counted in each stack on the ImageJ software. Statistical analysis of Sox10-positive cell counts in mutants and controls was carried out using a Mann-Whitney U test.

## NG2, PECAM, and Tuj1-positive surface area quantification

NG2-positive surface area quantifications in mutant $Egr2^{Cre/GFP(DT)},Rosa26R^{Tom}$ (n=4) and control (n=3) mice were performed on whole-mount skin preparations at E15.5, labeled for PECAM and NG2. For each embryo, three (controls) or six (mutants) z-stacks of different and non-overlapping fields of view were acquired using a Leica TCS SP5 (mutants) or Zeiss LSM 900 (controls) laser-scanning confocal microscope. The scanned surface area per field corresponds to 0.10 µm$^2$. A similar approach was used for Tuj1-positive surface area quantifications in mutant $Egr2^{Cre/GFP(DT)},Rosa26R^{Tom}$ (n=2) and control (n=2) mice, which were also performed on whole-mount skin preparations at E15.5, labeled for Sox10 and Tuj1. For each embryo, three (controls) or four (mutants) z-stacks of different and non-overlapping fields of view were acquired using a Zeiss LSM 900 laser-scanning confocal microscope. The algorithm to determine the area coverage of the NG2, PECAM, and Tuj1 signals was implemented in Python using the OpenCV (https://opencv.org/) library. Briefly, for both channels, a maximum z-projection of each stack was generated to simplify the subsequent analysis. Images were then pre-processed using histogram equalization, only for the PECAM channel, and Gaussian blurring to respectively enhance contrast and reduce high-frequency noise. They were then thresholded using an adaptive mean filter with a block size of 101 pixels for PECAM and 501 pixels for Tuj1 and NG2. Small objects were removed based on their area size (inferior to 300, 100, and 50 pixels for PECAM, Tuj1, and NG2, respectively) to clean up the masks. Finally, the percentage of coverage of each channel was

defined as the number of positive pixels in the mask divided by the total number of pixels in the image multiplied by 100. Statistical analyses of the PECAM, NG2, and Tuj1-positive surface areas between controls and mutants were carried out using a Mann-Whitney U test.

## Statistical analyses

Statistical analyses were carried out using a Mann-Whitney U test using the R software version 3.6.1 (http://www.r-project.org). Non-significant p-values are marked 'ns'. p-Values considered significant are indicated by asterisks as follows: *, p<0.05; **, p<0.01; ***, p<0.001. Data are represented as mean values ± standard deviation.

## Semi-quantitative RT-PCR

Total RNA (100 ng) was isolated from E12.5 embryonic skin, reverse transcribed using the pSuperscript III RNAse H reverse transcriptase (Invitrogen), and a mix of oligo-dT and random primers (Invitrogen), according to the manufacturer's instructions. PCR was performed as follows: 2 min at 94°C; 35 cycles of 2 min at 94°C, 1 min at 58°C, 30 s at 72°C. *Egr2* primer sequences were the following: (5'–3') GCAGAAGGAACGGAAGAGC; (3'–5') ACTGGTGTGTCAGCCAGAGC.

## Skin dissection and FACS

E12.5 *Egr2^Cre/+^,Rosa26R^Tom^* embryos were identified on the basis of Tomato expression under fluorescent stereomicroscope (Leica, Nussloch, Germany), and their head and viscera were removed. The back skin was dissected and then digested with collagenase/dispase type I (Merck/Roche) for 15 min at 37°C. Digestion was stopped by addition of 0.1 ml of fetal calf serum. Skin samples were then mechanically dissociated and the cell suspension was filtered. Dissociated cells were then resuspended in PBS, 1% BSA, and subjected to FACS. Tomato-positive cells were isolated, while dead cells and doublets were excluded by gating on a forward-scatter and side-scatter area versus width. Log RFP fluorescence was acquired through a 530/30 nm band pass. Internal Tomato-negative cells served as negative controls for FACS gating. Tomato-positive cells were sorted directly into lysis buffer for RT-PCR, or into PBS, 0.04% BSA for scRNA-seq experiments. To evaluate the purity of sorted cells, aliquots of the positive and negative fractions were sorted by FACS again with similar gating parameters, seeded onto coverslips and analyzed by immunohistochemistry with an anti-Tomato antibody.

## scRNA-seq and data analysis

Tomato-positive cells were isolated by FACS from E12.5 embryonic skin. Around 10,000 cells were loaded into one channel of the Chromium system using the V3 single-cell reagent kit (10X Genomics). Following capture and lysis, cDNAs were synthesized, then amplified by PCR for 12 cycles as per the manufacturer's protocol (10X Genomics). The amplified cDNAs were used to generate Illumina sequencing libraries that were each sequenced on one flow cell NextSeq500 Illumina. Cell Ranger v3.0.2 (10X Genomics) was used to process raw sequencing data. This pipeline converted Illumina base call files into Fastq format, aligned sequencing reads to a custom mm10 transcriptome with added sequences for the Tomato and Cre transgenes using the STAR aligner (*Li et al., 2009*), and quantified the expression of transcripts in each cell using Chromium barcodes. The count matrix generated by Cell Ranger v3.0.2 was loaded into R, thanks to the Seurat package v3.1.2 (*Stuart et al., 2019*) for R 3.6.1 (http://www.r-project.org). Empty droplets were filtered out using the EmptyDrops method implemented into the droplet Utils package v1.6.1 (*Lun et al., 2019*), with an FDR-adjusted p-value cut-off of 1E-03. Cells were selected if they expressed at least 500 genes including Tom, and if mitochondrial genes accounted for at most 20% of the total, and ribosomal genes for at least 7.5% of the total. Genes expressed in less than 10 cells were filtered out. Doublet cells were identified by a combination of two methods: the hybrid method implemented in the scds v1.2.0 package (*Bais et al., 2020*) and the method from the scDblFinder v1.0.0 package (*McGinnis et al., 2019*). This resulted in a dataset of 2672 single-cell transcriptomes, among which three contaminating cell types were detected: Myod1-positive myoblasts (n=64), PECAM-positive endothelial cells (n=22), Neurod1-positive sensory neurons (n=5), and C1qb-positive immune cells (n=3). These cell types, accounting for 3% of the total cell number, were removed from the raw dataset for the following analysis. Cells for which no counts were measured for Tomato (n=377) were also removed. Cell cycle scores and class prediction were performed with the cyclone method implemented in the scran v1.14.5 package

(*Lun et al., 2016*). Count values were normalized using the LogNormalize (*Hafemeister and Satija, 2019*) method implemented in Seurat, regressing out cell cycle (S minus G2M) as covariates. Principal component analysis (PCA)-based dimensionality reduction, graph-based cell clustering, and uniform manifold approximation and projection embedding were performed with Seurat, using 19 PCA dimensions with a 0.9 resolution and all the other parameters to default values.

## Materials availability

Mouse strains carrying the *Egr2^Cre* and *Rosa26R^Tom* alleles are freely available from The Jacksons Laboratory (USA). *Prss56^Cre* mice will require an MTA for future users.

## Acknowledgements

We are grateful to the IBENS animal, Imaging and Genomic facilities. The IBENS imaging facility is a member of the national infrastructure of France-BioImaging, supported by the ANR (ANR-10-INBS-04, ANR-10-LABX-54 MEMO LIFE, ANR-11-IDEX-0001-02 PSL 'Investments for the future') and by the 'Région Ile-de-France' (NERF N°2011-45, DIM Cerveau et Pensée 'Alpins'). The IBENS Genomic Facility was supported by France Génomique, managed by the ANR (ANR-10-INBS-09). Funding: The Charnay laboratory was financed by INSERM, CNRS, MESRI, and INCa. It has received support under the program 'Investissements d'Avenir' with the references: ANR-10-LABX-54 MEMOLIFE and ANR-11-IDEX-0001-02 PSL* Research University.

## Additional information

### Funding

| Funder | Grant reference number | Author |
| --- | --- | --- |
| Agence Nationale de la Recherche | ANR-10-LABX-54 MEMOLIFE | Patrick Charnay |
| Agence Nationale de la Recherche | ANR-11-IDEX-0001-02 PSL* Research University | Patrick Charnay |
| Institut National de la Santé et de la Recherche Médicale | | Patrick Charnay Piotr Topilko |
| Centre National de la Recherche Scientifique | | Patrick Charnay Piotr Topilko |
| Institut National Du Cancer | | Patrick Charnay Piotr Topilko |
| Ministère de l'Enseignement Supérieur et de la Recherche Scientifique | | Patrick Charnay Piotr Topilko |

The funders had no role in study design, data collection and interpretation, or the decision to submit the work for publication.

### Author contributions

Gaspard Gerschenfeld, Conceptualization, Data curation, Software, Formal analysis, Investigation, Writing - original draft; Fanny Coulpier, Conceptualization, Data curation, Software, Formal analysis, Investigation, Methodology, Project administration, Writing – review and editing; Aurélie Gresset, Pernelle Pulh, Investigation; Bastien Job, Data curation, Software, Formal analysis, Investigation; Thomas Topilko, Software, Formal analysis, Investigation; Julie Siegenthaler, Maria Eleni Kastriti, Investigation, Writing – review and editing; Isabelle Brunet, Resources, Writing – review and editing; Patrick Charnay, Supervision, Funding acquisition, Writing – review and editing; Piotr Topilko, Conceptualization, Supervision, Funding acquisition, Investigation, Methodology, Writing - original draft, Writing – review and editing

## Author ORCIDs
Gaspard Gerschenfeld (iD) http://orcid.org/0000-0002-2456-704X
Fanny Coulpier (iD) http://orcid.org/0009-0007-7638-3676
Isabelle Brunet (iD) http://orcid.org/0000-0002-5490-2937
Piotr Topilko (iD) http://orcid.org/0000-0001-7381-6770

## Ethics

All animal manipulations were performed according to French and European Union regulations. According to these regulations, no ethics committee approval was required for this study which only used mouse embryos and newborns.

## Decision letter and Author response

Decision letter https://doi.org/10.7554/eLife.69413.sa1
Author response https://doi.org/10.7554/eLife.69413.sa2

---

# Additional files

## Supplementary files
• Transparent reporting form

## Data availability

Single-cell RNA-seq data have been deposited in the ArrayExpress database at EMBL-EBI under accession number E-MTAB-8972.

The following dataset was generated:

| Author(s) | Year | Dataset title | Dataset URL | Database and Identifier |
|---|---|---|---|---|
| Gerschenfeld G, Coulpier F, Gresset-Hurpin A, Pulh P, Job B, Topilko T, Siegenthaler J, Kastriti ME, Brunet I, Charnay P, Topilko P | 2023 | Boundary caps: a major source of skin mural cells | https://www.ebi.ac.uk/arrayexpress/experiments/E-MTAB-8972/ | ArrayExpress, E-MTAB-8972 |

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
