## [Editor Report]

The authors show that Krox20 positive boundary cap cells travel along the nerves to the dermis and become incorporated into the vascular plexus to form dermal mural cells, confirmed by a cluster in single cell RNA sequencing. This provides the first evidence of a boundary cap contribution to a portion of mural cells in the trunk dermis.

---

## [Decision Letter]

**Decision letter after peer review:**

Thank you for submitting your article "Neural tube-associated boundary caps are a major source of mural cells in the skin" for consideration by *eLife*. Your article has been reviewed by 3 peer reviewers, and the evaluation has been overseen by a Reviewing Editor and Marianne Bronner as the Senior Editor. The following individuals involved in review of your submission have agreed to reveal their identity: Igor Adameyko (Reviewer #1); Tatiana Solovieva (Reviewer #2).

Essential revisions:

The reviewers find the paper potentially very interesting but also raise a major important question regarding the neural crest origin or lack thereof of these perivascular cells.

1) It is essential to provide further data regarding whether or not the boundary cap cells producing mural cells are of neural crest origin, as promised. To this end, you should perform additional lineage tracing with a Sox10-Cre line since boundary cap cells are Sox10 and Krox20 double positive. Wnt1-Cre would also be an appropriate line to use.

2) If the lineage tracing does not label mural cells, then the conclusion that these are neural crest derived should be revised and ideally it would be nice to show where they come from.

The full reviews are included below to help with your revision.

*Reviewer #1 (Recommendations for the authors):*

The manuscript by Gerschenfeld and co-authors is an interesting study focused around the question of a potential contribution from the trunk neural crest cells to the lineages of perivascular cells. I really enjoyed reading it, and I believe this question deserves a serious attention. Although the authors performed a lot of accurate experiments, the current results do not come together with the current knowledge in the field, mostly because other Cre strains recombining in the neural crest populations do not recapitulate trunk perivascular cells, which the authors find to be traced with Krox20 Cre line.

The key would be to rule out the neural crest (classical Sox10+/Wnt1+) or non-neural crest origin of these perivascular cells. It might be that the embryonic nerves contain non-neural crest-derived cells with the plasticity discovered by the authors.

Below I list my suggestions for the critical improvements:

1. Other NCC and SCP-specific Cre lines should be used in parallel. The authors shall try Sox10-CreERT2, PLP1-CReERT2, Wnt1-Cre or similar.

2. The authors should check the expression of Krox20-driven Cre mRNA throughout all investigated stages for ruling out any ectopic recombination.

3. The in situ for Krox20 in the entire embryo and mostly in skin should be done for addressing possible expression outside of the boundary cap.

4. In the case, where the authors will not be able to confirm the neural crest-derived nature of the pervascular cells in the trunk, the authors will need to re-write the manuscript suggesting an alternative origin. Indeed, it might be possible that ventral boundary cap is a heterogeneous populations, and is only partly derived from the classic neural crest. The rest of this population, in theory, might come from the ventral neural tube (*Sox2* Cre lines should clarify it) or from mesoderm (feels unlikely), and in that case Mesp1 Cre or similar might be helpful. Of note: the paper will appear even more interesting to me in case when non-neural crest progenitors are associated with the peripheral nerves and give rise to the perivascular component.

5. The transitions should be validated in single cell data with RNA velocity. Krox20 expression needs to be shown on UMAPs. It is well known that Schwann cell precursors give rise to endoneurial fibroblasts. Those can be similar in their transcriptomics profile to the collected perivascular populations, leading to the false idea of connectedness. This needs to be investigated deeper.

*Reviewer #2 (Recommendations for the authors):*

Krox20 is a marker for boundary cap (BC) cells. The authors trace BC derivatives in the mouse using Krox20Cre/+, Rosa26RTom embryos, where Tom-positive cells represent Krox20 derivatives. The authors identify BC derivatives that migrate along nerves towards the skin where a large proportion of these then detach from the nerves and instead attach to blood vessels. While these BC derived cells express glial markers for Schwann cell precursors (SCPs) (Sox10, BLBP) during their migration along the nerves, expression of these markers is reduced in some cells when they reach the skin. BC derived cells that detach from the nerves and instead associate with vessels express markers of mural cells (ABCC9, NG2, PDGFR β, SMA). At postnatal day 1, two-thirds of NG2-positive cells in the skin were Tom-positive, suggesting that two thirds of mural cells were BC-derived at this stage. Analysis of cell transcriptomes (from scRNAseq) of Tom-positive cells, at the stage when these cells begin to detach from the nerve (E12.5), revealed four general cell clusters; SCP-like, mesenchymal progenitor-like, transition-like cells, and a mural cell-like cluster. Inferred lineages from this data were used to suggest that SCPs go through a transition phase; from SCPs to mesenchymal progenitor like cells, before becoming mural cells.

Beautiful immunohistochemistry images provide strong support for the authors' claims through most of the paper. However, there are a couple more images that would be nice to have, to visualise some of the conclusions that authors derived from the scRNAseq data.

1. Authors say that for some cells, the transition from SCP to mural identity initiates while the cell is still attached to the nerve. The authors show antibody staining for SCP markers but not for mural cell markers at these stages (E12.5-E13.5, Figure 2). Antibody staining for both SCP and mesenchymal progenitor and/or mural cell markers in the same section showing positive signal for both in the same cell while detaching from a nerve would reinforce and visualise the authors' claims from the scRNAseq data.

2. Analysis of the scRNAseq data lead authors to suggest that BC derivatives also give rise to chondroprogenitors in the trunk. Antibody staining for chondroprogenitor markers (e.g. *Sox9*, Dlx5, Col2a1, *Fgfr2*) in the traced (Tom-positive) cells would reinforce the authors' claims from the inferred lineages from scRNAseq.

This is a great paper. My only recommendation to the authors would be to add a couple of antibody stainings. The most crucial would be for both SCP markers and mesenchymal progenitor and/or mural cell markers in the same section at E12.5, showing Tom-positive cells that co-express glial and mesenchymal and/or mural molecular markers in the same cell. Showing this in a Tom-positive cell that is detaching from a nerve would reinforce the authors' claims from the scRNAseq data for a 'transition' phase.

*Reviewer #3 (Recommendations for the authors):*

Gerschenfeld et al. investigate offspring of boundary cap (BC) cells in the embryonic dermis using lineage tracing in mice. They report that Krox20 positive BC cells travel along the nerves to the dermis, where they undergo a glial to vascular identity switch, detach from nerves and incorporate into the vascular plexus to become mural cells. Knockout of Krox20 does not affect cell migration, suggesting this gene is a marker but not involved in cell fate switch. Quantifications of immune-labelled sections showed that up to 60% of neonatal dermal mural cells are derived from Krox20+ BC precursors. Single cell RNA sequencing confirms heterogeneity of Krox20+ cells, including a cluster of cells with mural cell identity.

Overall, this manuscript is well written and illustrated and provides what constitutes to my knowledge the first evidence of a neural crest origin of a portion of mural cells in the dermis of the trunk, thereby adding to the multiple known origins of vascular mural cells.

I have just a few comments that should be addressed before publication can be recommended.

Figure 3 Quantification of PdgfrB+Tom+ double positive cells would strengthen the quantification and the authors conclusion and should be added.

Figure 3 and S5a-c show that Tom-positive MCs do not form a continuum and are separated by non-traced MCs, suggesting that multiple sources of mural cells contribute to the dermal vessel wall. Why do the authors think that is?

Figure 4 and 5: can the scRNA seq analysis be further exploited towards more mechanistic understanding of the transdifferentiation. It would also be interesting to compare mural cells from different origins ie BC derived versus other sources.

Figure 6 Ablation of Krox20+ cells using DTR led to embryonic lethality, and analysis of a few surviving embryos revealed incomplete ablation of BC progeny in 4/6 embryos, while two showed compete loss of BCs and were chosen for analysis of the vascular network. The authors measured vascular area in those embryos and found no differences. This result is surprising and deserves some consideration. One would expect lack of mural cell coverage to lead to microaneurysms and hemorrhage, yet this was apparently not observed? The vasculature in Figure 6 panels C and F does look different between controls and mutants, I would encourage the authors to further characterize the vascular defects in these mice. Also, presence and absence of Schwann cells in these mice should be shown and discussed.

Figure S2c: please label the two circular NG2+ structures.

[Editors’ note: further revisions were suggested prior to acceptance, as described below.]

Thank you for resubmitting your work entitled "Neural tube-associated boundary caps are a major source of mural cells in the skin" for further consideration by *eLife*. Your revised article has been evaluated by Marianne Bronner (Senior Editor) and a Reviewing Editor.

The manuscript has been improved but there are a few remaining issues that need to be addressed, as outlined below:

One of the main concerns from the original submission was the origin of the perivascular cells. The authors have addressed this and tested for possible neural crest (Sox10Cre-Ert2 and Wnt1Cre lines), neural tube (Olig2Cre line), and pial (Tbx18Cre-Ert2 line) origins. Sox10Cre-Ert2, Wnt1Cre lines, and Olig2Cre lines did not give rise to traced mural cells. With the Tbx18Cre-Ert2 line, tamoxifen was administered to pregnant females at E9.5, and embryos analyzed at E11.5. SOX10 negative traced cells were identified at E11.5 along the ventral root and spinal nerve. This was used to suggest a pial cell origin of perivascular cells. If possible, it would be beneficial to have a visual of the expression of Tbx18 at E9.5, to visualize the origin of the traced cells. It is also unclear why traced cells from the TBX18 line were not followed through to later stages (E14.5) where cells could be more stringently verified as mural cells by immunostaining for some mural cell markers (as in Figure 2A-D) and their morphology and migratory behavior could also be better verified as 'mural cell'.

---

## [Author Response]

Essential revisions:The reviewers find the paper potentially very interesting but also raise a major important question regarding the neural crest origin or lack thereof of these perivascular cells.1) It is essential to provide further data regarding whether or not the boundary cap cells producing mural cells are of neural crest origin, as promised. To this end, you should perform additional lineage tracing with a Sox10-Cre line since boundary cap cells are Sox10 and Krox20 double positive. Wnt1-Cre would also be an appropriate line to use.

As indicated in detail below in the specific responses to the referees, we have performed additional tracing experiments designed to investigate the origin of the perivascular cells. Using Sox10^Cre-Ert2^ and Wnt1^Cre^ lines, as well as a neural tube tracing line (Olig2^Cre^), we show that the ventral BC subpopulation that gives rise to mural cells does not originate from the neural crest or from the neural tube.

Our new findings are detailed in a new subsection in the Results on page 8 (lines 1 to 28), with its associated Figure 4. The Discussion was also substantially modified to reflect these findings from pages 12 to 14. The Methods were also updated on page 15 (lines 2 to 12).

2) If the lineage tracing does not label mural cells, then the conclusion that these are neural crest derived should be revised and ideally it would be nice to show where they come from.

This is indeed what we found. On this basis, we have chosen to explore whether the developing pia matter could contribute to BCs. Using a Tbx18^Cre-Ert2^ line, we show that traced cells are found in close contact with the ventral root, and that they migrate along the nerves on top of Schwann cell precursors. In addition, we performed RNA scope in situ hybridization and show that some Krox20-positive BC cell derivatives along the ventral root do not express Sox10, but are positive for mesenchymal markers such as Tbx18, Col1a1 and Pdgfra. Finally, after upgrading our scRNA-seq analysis pipeline, we observe two separate superclusters of Tomato-positive cells with glial and mesenchymal signatures, further supporting a dual origin of ventral BCs, with a neural crest-derived population at the origin of SCPs and a mesenchymal-like population at the origin of mural cells.

Our new findings are detailed in two new subsections in the Results from page 6 (line 30) to page 10 (line 5), with their associated Figures 3 to 5. The Discussion was also substantially modified to reflect these findings from page 12 to 14. The Methods were updated on page 16 (lines 1 to 17) and from page 19 (line 8) to 20 (line 5).

Reviewer #1 (Recommendations for the authors):The manuscript by Gerschenfeld and co-authors is an interesting study focused around the question of a potential contribution from the trunk neural crest cells to the lineages of perivascular cells. I really enjoyed reading it, and I believe this question deserves a serious attention. Although the authors performed a lot of accurate experiments, the current results do not come together with the current knowledge in the field, mostly because other Cre strains recombining in the neural crest populations do not recapitulate trunk perivascular cells, which the authors find to be traced with Krox20 Cre line.The key would be to rule out the neural crest (classical Sox10+/Wnt1+) or non-neural crest origin of these perivascular cells. It might be that the embryonic nerves contain non-neural crest-derived cells with the plasticity discovered by the authors.Below I list my suggestions for the critical improvements:1. Other NCC and SCP-specific Cre lines should be used in parallel. The authors shall try Sox10-CreERT2, PLP1-CReERT2, Wnt1-Cre or similar.

We have performed additional lineage tracing studies using neural crest (Sox10^Cre-Ert2^ and Wnt1^Cre^) as well as neural tube (Olig2^Cre^) tracing lines. None of them labeled mural cells in the skin indicating that ventral BC subpopulation at their origin does not derive from the neural crest or neural tube.

Our new findings are detailed in a new subsection in the Results on page 8 (lines 1 to 28), with its associated Figure 4. The Discussion was also substantially modified to reflect these findings from pages 12 to 14. The Methods were also updated on page 15 (lines 2 to 12).

2. The authors should check the expression of Krox20-driven Cre mRNA throughout all investigated stages for ruling out any ectopic recombination.

We agree that exploring possible ectopic Cre recombination is of importance. To investigate this issue, we attempted to reveal Cre mRNA using RNAscope and Cre protein with immunostainings. However, both techniques are known to be delicate with Cre and this actually did not work. To tackle the issue in another way, we included the Cre sequence in our scRNA-seq analysis. We did not observe any *Cre* expression either at E12.5 (in our published dataset, which included 2209 cells), or at E13.5, using another unpublished dataset (1029 cells, data not shown). Although this analysis does not cover earlier stages, it demonstrates that from E12.5 no ectopic expression can be detected.

We have mentioned the absence of Cre expression in the scRNA-seq dataset in the Results on page 9 (lines 5 to 7).

3. The in situ for Krox20 in the entire embryo and mostly in skin should be done for addressing possible expression outside of the boundary cap.

This constitutes a related, important issue and we have performed additional analyses on ventral roots and the skin from E12.5 embryos. We have found that at that stage *Krox20* expression is restricted to BCs and is totally absent from skin. These data reinforce our previous observations obtained by in situ hybridization and RT-PCR data to exclude the possibility of *Krox20* expression in other sites than BCs.

Our new findings are detailed in the first subsection in the Results on page 5 (lines 14 to 20), with its associated Figure 1—figure supplement 1 (C-J). The Methods were also updated on page 16 (lines 1 to 17).

4. In the case, where the authors will not be able to confirm the neural crest-derived nature of the pervascular cells in the trunk, the authors will need to re-write the manuscript suggesting an alternative origin. Indeed, it might be possible that ventral boundary cap is a heterogeneous populations, and is only partly derived from the classic neural crest. The rest of this population, in theory, might come from the ventral neural tube (Sox2 Cre lines should clarify it) or from mesoderm (feels unlikely), and in that case Mesp1 Cre or similar might be helpful. Of note: the paper will appear even more interesting to me in case when non-neural crest progenitors are associated with the peripheral nerves and give rise to the perivascular component.

We have explored whether the neural tube or the developing pia matter (future meninges) might contribute to the ventral to BC cells population. Using an *Olig2^Cre^* line, we have found no traced cells in the periphery, excluding a neural tube contribution. In contrast, using a *Tbx18^Cre-Ert2^* line, we have identified traced cells on the ventral roots, within the BCs and migrating along the nerves on top of Schwann cell precursors. In addition, we performed RNAscope in situ hybridization analyses that revealed the presence on the ventral root of *Krox20*-positive BC derivatives negative for Sox10 and displaying mesenchymal markers such as Tbx18, Col1a1 and Pdgfra. Finally, upon upgrading our scRNA-seq analysis pipeline we have observed two distinct superclusters of Tomatopositive cells that further support the idea that ventral BCs are composed of two subpopulations: neural crest-derived cells that give rise to SCPs and cells with a mesenchymal identity that will give rise to mural cells and may be pia matter-derived.

Our new findings are detailed in two new subsections in the Results from page 6 (line 30) to page 10 (line 5), with their associated Figures 3 to 5. The Discussion was also substantially modified to reflect these findings from page 12 to 14. The Methods were updated on page 16 (lines 1 to 17) and from page 19 (line 8) to 20 (line 5).

5. The transitions should be validated in single cell data with RNA velocity. Krox20 expression needs to be shown on UMAPs. It is well known that Schwann cell precursors give rise to endoneurial fibroblasts. Those can be similar in their transcriptomics profile to the collected perivascular populations, leading to the false idea of connectedness. This needs to be investigated deeper.

To address this issue, we have upgraded our scRNA-seq analysis pipeline and found that most of the Tom-positive cells that connected the two super-clusters did not reach a proper quality and should be excluded from the analysis. In these conditions, it appears that the two super-clusters are clearly separated, which is consistent with the observations presented above. Together our data support the existence of a major heterogeneity in ventral BCs, both in terms of identity and origin as discussed above.

Our new scRNA-seq analysis is detailed in the Results from page 8 (line 29) to page 10 (line 5), with its associated Figure 5 and Figure 5—figure supplement 1. The Discussion was also substantially modified to reflect these findings from page 12 to 14. The Methods were also updated from page 19 (line 8) to 20 (line 5).

Reviewer #2 (Recommendations for the authors):Krox20 is a marker for boundary cap (BC) cells. The authors trace BC derivatives in the mouse using Krox20Cre/+, Rosa26RTom embryos, where Tom-positive cells represent Krox20 derivatives. The authors identify BC derivatives that migrate along nerves towards the skin where a large proportion of these then detach from the nerves and instead attach to blood vessels. While these BC derived cells express glial markers for Schwann cell precursors (SCPs) (Sox10, BLBP) during their migration along the nerves, expression of these markers is reduced in some cells when they reach the skin. BC derived cells that detach from the nerves and instead associate with vessels express markers of mural cells (ABCC9, NG2, PDGFR β, SMA). At postnatal day 1, two-thirds of NG2-positive cells in the skin were Tom-positive, suggesting that two thirds of mural cells were BC-derived at this stage. Analysis of cell transcriptomes (from scRNAseq) of Tom-positive cells, at the stage when these cells begin to detach from the nerve (E12.5), revealed four general cell clusters; SCP-like, mesenchymal progenitor-like, transition-like cells, and a mural cell-like cluster. Inferred lineages from this data were used to suggest that SCPs go through a transition phase; from SCPs to mesenchymal progenitor like cells, before becoming mural cells.Beautiful immunohistochemistry images provide strong support for the authors' claims through most of the paper. However, there are a couple more images that would be nice to have, to visualise some of the conclusions that authors derived from the scRNAseq data.

We thank the referee for his/her encouraging comments. Please note that the manuscript has been substantially modified following additional analyses detailed in the answers to the editor and to the other referees.

1. Authors say that for some cells, the transition from SCP to mural identity initiates while the cell is still attached to the nerve. The authors show antibody staining for SCP markers but not for mural cell markers at these stages (E12.5-E13.5, Figure 2). Antibody staining for both SCP and mesenchymal progenitor and/or mural cell markers in the same section showing positive signal for both in the same cell while detaching from a nerve would reinforce and visualise the authors' claims from the scRNAseq data.

To look for nerve-attached cells with SCP or mesenchymal identities, we have performed immunolabeling and RNAscope in situ hybridization analyses on Krox20-traced ventral roots and subcutaneous nerves. Using neural crest/glial (Sox10) and mesenchymal (Tbx18, Col1a1 and Pdgfra) probes, we have identified traced cells on ventral roots that do not express Sox10 and also observed traced cells on nerves that express the mesenchymal markers. On subcutaneous nerves, we have identified Krox20-traced Sox10-negative cells. Although we could not perform double-labeling analyses, these data strongly support the presence on nerves of two subpopulations of BC-derived cells with glial and mesenchymal identities.

Our new findings are detailed in a new subsection in the Results from page 6 (line 30) to page 7 (line 34), with its associated Figure 3. The Discussion was also substantially modified to reflect these findings from page 12 to 14. The Methods were updated on page 16 (lines 1 to 17).

2. Analysis of the scRNAseq data lead authors to suggest that BC derivatives also give rise to chondroprogenitors in the trunk. Antibody staining for chondroprogenitor markers (e.g. Sox9, Dlx5, Col2a1, Fgfr2) in the traced (Tom-positive) cells would reinforce the authors' claims from the inferred lineages from scRNAseq.

We agree that immunostainings would be potentially of interest to support the scRNAseq data. However, given the known expression of *Krox20* at E14 in chondroprogenitors, we felt that the time window to perform such an analysis was too narrow and that additional controls would be required to exclude late ectopic *Krox20* activation. This did not appear to be feasible in a reasonable time, as we had to focus on other aspects more central to the major messages of the paper.

Reviewer #3 (Recommendations for the authors):Gerschenfeld et al. investigate offspring of boundary cap (BC) cells in the embryonic dermis using lineage tracing in mice. They report that Krox20 positive BC cells travel along the nerves to the dermis, where they undergo a glial to vascular identity switch, detach from nerves and incorporate into the vascular plexus to become mural cells. Knockout of Krox20 does not affect cell migration, suggesting this gene is a marker but not involved in cell fate switch. Quantifications of immune-labelled sections showed that up to 60% of neonatal dermal mural cells are derived from Krox20+ BC precursors. Single cell RNA sequencing confirms heterogeneity of Krox20+ cells, including a cluster of cells with mural cell identity.Overall, this manuscript is well written and illustrated and provides what constitutes to my knowledge the first evidence of a neural crest origin of a portion of mural cells in the dermis of the trunk, thereby adding to the multiple known origins of vascular mural cells.I have just a few comments that should be addressed before publication can be recommended.Figure 3 Quantification of PdgfrB+Tom+ double positive cells would strengthen the quantification and the authors conclusion and should be added.

We agree with the reviewer that such a quantification would be of interest. However, it is not technically feasible on wholemount immunostaining. Pdgfrb immunostaining presents a higher background noise and is more difficult to attribute to a specific cell than NG2, for which we have performed quantifications, which were already challenging. An alternative way would be to use flow-cytometry, but it would be also challenging to restrict the analysis to cells attached to the vasculature.

Figure 3 and S5a-c show that Tom-positive MCs do not form a continuum and are separated by non-traced MCs, suggesting that multiple sources of mural cells contribute to the dermal vessel wall. Why do the authors think that is?

Our study and the one from Yamazaki and colleagues suggest a dual origin for mural cells in the skin vasculature. Approximately one-third of them originate from myeloid progenitors and the remaining ones derive from a subpopulation of ventral root BCs. The fact that each model of mural cell tracing shows a non continuous distribution of traced cells along blood vessels, while we know that the distribution of mural cells is homogeneous and continuous suggest that the two populations are intermingled. This might results from a simultaneous colonization of the vascular plexus by both populations. Interestingly, the genetic ablation of one population leads only to a partial compensation by the cells from the other origin.

We have updated the Discussion to address this question on page 12 (lines 25 to 30).

Figure 4 and 5: can the scRNA seq analysis be further exploited towards more mechanistic understanding of the transdifferentiation. It would also be interesting to compare mural cells from different origins ie BC derived versus other sources.

Additional analyses were performed and, as mentioned above, strongly suggest the existence of at least two subpopulations of ventral BCs with glial and mesenchymal characteristics instead of bi-potent glial progenitors migrating along the nerves to the skin and changing their fate into mural derivatives.

Our new scRNA-seq analysis is detailed in the Results from page 8 (line 29) to page 10 (line 5), with its associated Figure 5 and Figure 5—figure supplement 1. The Discussion was also substantially modified to reflect these findings from page 12 to 14. The Methods were also updated from page 19 (line 8) to 20 (line 5).

Whether the dual origin of trunk skin mural cells is accompanied by molecular and especially functional differences is indeed a primary question. However, this is a topic in itself that has not been explored in the present study.

Figure 6 Ablation of Krox20+ cells using DTR led to embryonic lethality, and analysis of a few surviving embryos revealed incomplete ablation of BC progeny in 4/6 embryos, while two showed compete loss of BCs and were chosen for analysis of the vascular network. The authors measured vascular area in those embryos and found no differences. This result is surprising and deserves some consideration. One would expect lack of mural cell coverage to lead to microaneurysms and hemorrhage, yet this was apparently not observed? The vasculature in Figure 6 panels C and F does look different between controls and mutants, I would encourage the authors to further characterize the vascular defects in these mice.

Regarding embryonic lethality at E15.5 following the targeted ablation of Krox20expressing BC cells, we strongly suspect that it is the ablation of the BC-derived MC component that is responsible for two reasons:

1. We have shown that targeted ablation of rhombomeres 3 and 5, where Krox20 is first expressed between E8.5 and E9.5, leads to death within two weeks after birth (SchneiderMaunoury et al., 1993);

2. We also know that the targeted ablation of neural crest-derived cells which express Krox20, by combining *Wnt1^Cre^* and *Krox20^GFP/DT^* mice, also leads to death within two weeks after birth (Odelin et al., 2018 and unpublished data).

In conclusion, while the accumulated evidence points to the role of BC-derived MCs, their implication remains to be formally demonstrated. We mention this point in the Discussion from page 13 (line 32) to page 14 (line 3).

We agree that the vasculature in panels C and F looks different. However, we have found that skin wholemount samples at the same stage can differ greatly in terms of aspect, even between littermates, likely because of factors not taken into account such as the exact skin location. Hence, caution is warranted when comparing these two pictures.

Overall, further investigations are needed to prove this hypothesis and study the vascular phenotype appropriately, but we felt that this would not be feasible in a reasonable time for two main reasons:

1. When these embryos are retrieved at E15.5 they usually are already moribund, which may induce post mortem changes in the vascular plexus that are not linked directly to the lack of MCs.

2. Generating embryos with complete ablation proved to be difficult with only 4 embryos on 60 collected embryos.

Also, presence and absence of Schwann cells in these mice should be shown and discussed.

Regarding the impact of the ablation of Krox20-expressing BCs on Schwann cells and nerve development, we have generated two additional mutants and performed immunostainings on them as well as two controls with neural (TuJ1) and glial (Sox10) markers. As a result, we did not observe any alteration in the innervation pattern of the skin nor the distribution of Schwann cell precursors along these nerves.

We have detailed this analysis in the relevant subsection in the Results on page 11 (lines 12 to 17), with its associated Figure 6—figure supplement 1. The Methods were also updated from page 17 (line 16) to page 18 (line 12).

Figure S2c: please label the two circular NG2+ structures.

These two circular structures are hair follicles. We have labelled them in panel C of Figure 2—figure supplement 1 of the Supplementary Materials.

[Editors’ note: what follows is the authors’ response to the second round of review.]

The manuscript has been improved but there are a few remaining issues that need to be addressed, as outlined below:One of the main concerns from the original submission was the origin of the perivascular cells. The authors have addressed this and tested for possible neural crest (Sox10Cre-Ert2 and Wnt1Cre lines), neural tube (Olig2Cre line), and pial (Tbx18Cre-Ert2 line) origins. Sox10Cre-Ert2, Wnt1Cre lines, and Olig2Cre lines did not give rise to traced mural cells. With the Tbx18Cre-Ert2 line, tamoxifen was administered to pregnant females at E9.5, and embryos analyzed at E11.5. SOX10 negative traced cells were identified at E11.5 along the ventral root and spinal nerve. This was used to suggest a pial cell origin of perivascular cells. If possible, it would be beneficial to have a visual of the expression of Tbx18 at E9.5, to visualize the origin of the traced cells.

To address the question about the origin of traced cells, we performed RNAscope *in*

*situ* hybridization with the Tbx18 probe on E9.5 and E10.5 embryos, as tamoxifen-induced recombination is considered to take around 12-24h to occur. On these sections, we found Tbx18-expressing cells located around the neural tube, dorsal roots and dorsal root ganglia, in the area that corresponds to the developing pia matter.

We have added a supplemental figure (Figure 4—figure supplement 1) that contains the

RNAscope data at E9.5 and E10.5 with its corresponding legend on page 29. We have also updated the Results section to mention these results on Page 8 lines 22 to 25.

“Furthermore, at E9.5, we detected Tbx18 expression in the vicinity of the neural tube, in a layer of mesenchymal progenitors that gives rise to fibroblasts in the meninges (Figure 4-figure supplement 1), consistent with a previous report (DeSisto et al., 2020).”

It is also unclear why traced cells from the TBX18 line were not followed through to later stages (E14.5) where cells could be more stringently verified as mural cells by immunostaining for some mural cell markers (as in Figure 2A-D) and their morphology and migratory behavior could also be better verified as 'mural cell'.

We agree that additional fate mapping of the Tbx18 lineage at a later stage was

necessary, and as suggested we performed immunostaining on sections of *Tbx18Cre-*

*Ert2/+,Rosa26RTom* embryos at E14.5 that had received tamoxifen at E9.5. Near the neural tube, we found numerous traced cells around blood vessels that expressed mural cell markers such as NG2 and SMA, as well as cells around and within the developing spinal nerve that did not express Sox10. In the skin, we also found that traced cells did not express Sox10 and that some of them were in contact with blood vessels and expressed NG2. Overall, although caution is warranted given the preliminary nature of our analyses on the Tbx18 line, our observations are in line with the hypothesis of a pial origin of mural cells.

We have added a supplemental figure (Figure 4—figure supplement 2) that contains the

immunostaining data at E14.5 with its corresponding legend on page 29. We have also

updated the Results section to mention these data on Page 8 lines 22 to 25:

“At E14.5, we observed numerous Tom-positive cells along blood vessels co-expressing NG2 or/and SMA (Figure 4—figure supplement 2A-F, J-L). None of them express Sox10 (Figure 4-figure supplement 2G-I, M-O).”